# Certified Robustness Training: Closed-Form Certificates via CROWN

## Abstract

Adversarial training reshapes neural network decision boundaries by pushing them away from adversarial examples, but this approach ignores a crucial geometric factor: the local curvature that determines how steeply network outputs change with input perturbations. We introduce a fundamentally different approach that optimizes certified robustness by directly reshaping decision boundary geometry during training. Our key insight is that CROWN's linear bounds encode both the safety margin and input sensitivity needed for closed-form certified radius computation, transforming expensive verification into efficient geometric analysis. We derive differentiable expressions that enable direct optimization of the margin-over-slope ratio underlying certified robustness, creating networks with inherently robust decision regions rather than boundaries hardened against specific attacks. Our hybrid training method combines adversarial training's broad coverage with geometric certified objectives applied to hard examples, achieving 98.33% clean accuracy and 71.1% certified robustness at $\epsilon = 0.03$ on MNIST—outperforming both PGD adversarial training (61.7%) and randomized smoothing (53.1%) in ReLU-based networks. On DC optimal power flow regression, we demonstrate controllable accuracy-safety trade-offs critical for engineering applications. By making certified robustness certificates both computationally tractable and differentiable, our approach enables robustness-aware learning that produces networks robust by geometric design rather than adversarial accident.

## 1 Introduction

Adversarial training has emerged as the dominant approach for learning robust neural networks, but it suffers from a fundamental limitation: it primarily moves decision boundaries without changing their local geometry. When a network encounters adversarial examples during training, gradient-based methods shift the boundary away from these threats while preserving the network's inherent sensitivity to input perturbations. This creates a cat-and-mouse dynamic where stronger attacks find new vulnerabilities in regions that remain geometrically fragile, leading to thin vulnerable slivers that evade detection during training but compromise robustness in deployment.

We propose a fundamentally different approach that addresses robustness at its geometric root: instead of merely pushing decision boundaries away from adversarial examples, we reshape their local curvature to create inherently more robust decision regions. Our key insight is that certified robustness bounds encode precise information about both the safety margin at a point and the network's input sensitivity—and optimizing these quantities directly during training leads to decision boundaries that are robust by construction rather than by adversarial hardening.

The central contribution of this work is showing that linear bound propagation methods like CROWN (Zhang et al., 2020b), widely used for post-hoc verification, can be transformed into differentiable training objectives that optimize certified radius bounds in closed form. Specifically, CROWN's affine bounds $\ell_s(x) = a_s^T x + \beta_s \leq s(x) \leq u_s(x) = \tilde{a}_s^T x + \tilde{\beta}_s$ naturally encode the geometric quantities needed for radius computation: the safety margin $m(c) = a_s^T c + \beta_s$ and the worst-case input sensitivity $\|a_s\|_1$. The certified radius becomes a simple margin-over-slope ratio: $r(c) = m(c)/\|a_s\|_1$.

This geometric perspective reveals why certified training succeeds where adversarial training struggles. While adversarial methods implicitly optimize margin through example-based learning, they

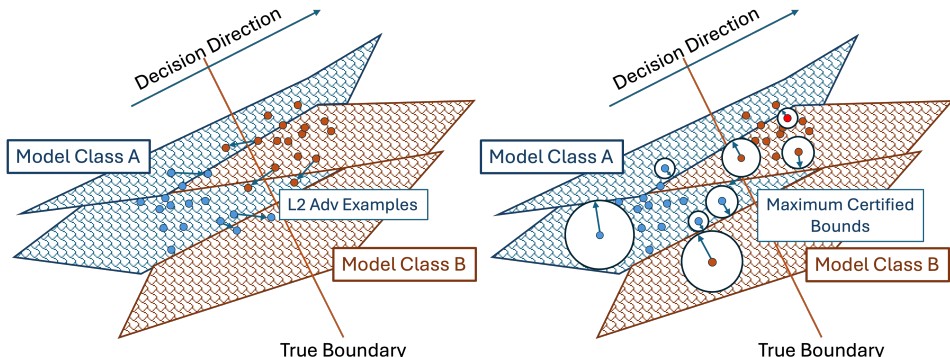

Figure 1: **Attack-only training moves the boundary; our certified objective changes its curvature (with $\ell_\infty$ radii).** Hatched polygons mark activation-stable (locally affine) regions of a ReLU network; colors denote classes. *Left:* adversarial/smoothing training largely translates the boundary along the decision direction, leaving thin vulnerable slivers. *Right:* adding the $\beta$-CROWN–driven radius loss enlarges per-sample certified $\ell_\infty$ balls (drawn as circles for readability) by simultaneously increasing margin and reducing input slope on neighboring facets, effectively reshaping local curvature and aligning better with the true boundary.

ignore the equally important sensitivity term. Our approach optimizes both simultaneously: increasing margin while reducing input sensitivity creates larger certified neighborhoods and fundamentally changes the local decision geometry. Figure 1 illustrates this difference—adversarial training translates boundaries, while our certified objectives reshape their curvature.

The practical implications are significant. Because CROWN coefficients are differentiable functions of network parameters, our certified radius bounds inherit this differentiability, enabling direct gradient-based optimization during training. This transforms expensive post-hoc verification into an efficient training signal that guides the network toward inherently robust representations.

Our hybrid training approach combines the broad coverage of adversarial training with the geometric precision of certified radius optimization. We apply adversarial training to all examples for baseline robustness, then selectively apply certified radius penalties to hard examples where geometric reshaping provides the greatest benefit. This selective approach balances computational cost with robustness gains while avoiding the optimization difficulties that can arise from applying strong certified constraints globally.

**Contributions and experimental validation.** We make certified robustness both theoretically principled and practically effective through four key contributions:

- **Closed-form certified radii:** We derive exact formulas converting CROWN bounds into certified radius expressions, enabling $O(d)$-time robustness assessment versus exponential MILP complexity.
- **Differentiable certified training:** We provide explicit matrix formulations showing how CROWN coefficients depend on network parameters, making certified radii fully differentiable for gradient-based optimization.
- **Geometric robustness insight:** We demonstrate that certified training reshapes decision boundary curvature rather than merely translating boundaries, addressing the fundamental limitations of adversarial training.
- **Superior empirical performance:** Our hybrid method achieves 98.33% clean accuracy and 71.1% certified robustness at $\epsilon = 0.03$ on MNIST, outperforming both PGD adversarial training (61.7%) and randomized smoothing (53.1%) on ReLU-based neural networks.

We validate our approach across two distinct domains: MNIST classification, where our method demonstrates clear improvements over established baselines in both clean accuracy and certified robustness; and DC-OPF power system control, where we show how certified violation penalties

enable principled navigation of the accuracy-safety trade-off critical in engineering applications. Together, these results demonstrate that optimizing certified geometry during training produces networks that are robust by design rather than by adversarial accident.

## 2 BACKGROUND: CERTIFIED BOUNDS AND LINEAR RELAXATIONS

Having motivated the geometric perspective on certified robustness, we now establish the mathematical foundation for our approach. We review how linear bound propagation methods compute affine bounds that encode the geometric structure of neural network decision boundaries.

**Network architecture and problem formulation.** We consider a feedforward neural network $f_\theta : \mathbb{R}^d \to \mathbb{R}^k$ with parameters $\theta = (W^{(1)}, b^{(1)}, \ldots, W^{(L)}, b^{(L)})$, where $L$ denotes the number of layers. For an input $x \in \mathbb{R}^d$, the network output is $f_\theta(x) = (f_1(x), \ldots, f_k(x))$.

For classification tasks, we focus on adversarial robustness: given a correctly classified input with true class $t \in \{1, \ldots, k\}$, we study pairwise logit margins $m_{t,j}(x) = f_t(x) - f_j(x)$ for $j \neq t$. The network maintains its prediction when all margins $m_{t,j}(x) > 0$. Without loss of generality, we can focus on bounding scalar quantities $s(x) \in \mathbb{R}$ derived from network outputs—for regression, $s(x)$ might represent constraint violations; for classification, it typically represents logit margins where maintaining $s(x) > 0$ ensures correct prediction. We omit explicit dependence on $\theta$ when clear from context.

**Input regions and perturbation models.** We work with compact regions in input space. The axis-aligned box is
$$[L, U] = \{x \in \mathbb{R}^d : \ L \leq x \leq U \text{ (elementwise)}\}.$$
Given a center $c \in \mathbb{R}^d$ and radius $r \geq 0$, the $\ell_\infty$-ball is
$$\mathcal{B}_\infty(c, r) = \{x \in \mathbb{R}^d : \ \|x - c\|_\infty \leq r\}.$$

We focus on $\ell_\infty$ perturbations as they naturally model pixel-wise bounded adversarial attacks in image domains and element-wise operational tolerances in control applications. Throughout, $[L, U]$ denotes a fixed domain, and we seek to certify balls $\mathcal{B}_\infty(c, r) \subseteq [L, U]$ around centers $c$.

**Linear bound propagation and certified affine bounds.** The key insight behind linear bound propagation methods like CROWN is to replace nonlinear activations with linear relaxations, enabling efficient bound computation through matrix operations. These methods provide certified affine bounds on scalar network quantities $s(x)$ that are valid uniformly over input boxes $[L, U]$ (Zhang et al., 2020b; Wang et al., 2021b).

Specifically, for any scalar function $s(x)$ induced by the network, CROWN computes affine functions:

$$\ell_s(x) = a_s^\top x + \beta_s, \tag{1}$$
$$u_s(x) = \tilde{a}_s^\top x + \tilde{\beta}_s, \tag{2}$$

where the coefficients $(a_s, \beta_s, \tilde{a}_s, \tilde{\beta}_s)$ depend on both the network parameters $\theta$ and the input domain $[L, U]$. These bounds satisfy the certified sandwich inequality

$$\ell_s(x) \ \leq \ s(x) \ \leq \ u_s(x) \qquad \text{for all } x \in [L, U].$$

The coefficients are computed through a backward pass that propagates linear bounds layer by layer, making the approach scalable to large networks. The bounds become tighter when the input region is smaller or when the network exhibits more stable activation patterns within that region.

**Computational efficiency and geometric structure.** CROWN bound computation requires only one forward and one backward pass through the network, scaling polynomially with network size. Crucially for our approach, the resulting affine bounds reveal the local geometric structure of the network's decision boundary within the specified region. The coefficient vectors $a_s$ and $\tilde{a}_s$ capture how the network output changes with input variations, while the offset terms $\beta_s$ and $\tilde{\beta}_s$ reflect the network's behavior at the reference point.

This geometric information, encoded directly in CROWN's affine bounds, will serve as the foundation for deriving closed-form expressions for certified robustness radii in the following sections.

# 3 FROM CROWN BOUNDS TO CLOSED-FORM RADII

The power of our geometric approach lies in transforming CROWN's linear bounds into explicit certified radius expressions. We begin by examining the computational challenges of exact verification, then show how CROWN bounds contain precisely the geometric information needed for closed-form radius computation.

## 3.1 THE COMPUTATIONAL CHALLENGE OF EXACT VERIFICATION

For a ReLU network $f_\theta : \mathbb{R}^d \to \mathbb{R}^k$ and scalar safety property $s(x) \in \mathbb{R}$ (e.g., classification margin $m_{t,j}(x) = f_t(x) - f_j(x)$), the exact certified radius at center $c$ is:

$$r^\star(c) = \min_{x \in \mathbb{R}^d} \|x - c\|_\infty \quad \text{subject to} \quad s(x) \le 0. \tag{3}$$

This optimization can be solved exactly using mixed-integer linear programming by encoding ReLU constraints with binary variables $\delta_i^\ell \in \{0, 1\}$ for each neuron (Fischetti & Jo, 2018; Tjeng et al., 2019; Bunel et al., 2018; Chehade et al., 2025):

$$h_i^\ell \ge z_i^\ell, \quad h_i^\ell \ge 0, \tag{4}$$
$$h_i^\ell \le z_i^\ell - L_i^\ell(1 - \delta_i^\ell), \quad h_i^\ell \le U_i^\ell \delta_i^\ell, \tag{5}$$

where $(L_i^\ell, U_i^\ell)$ are pre-computed activation bounds.

While MILP formulations provide exact solutions, they suffer from fundamental limitations: exponential worst-case complexity in the number of neurons, sensitivity to activation bound tightness, and incompatibility with gradient-based optimization due to discrete variables. These constraints motivate our search for tractable approximations that preserve geometric insight.

## 3.2 THE GEOMETRIC STRUCTURE IN CROWN BOUNDS

CROWN (Zhang et al., 2020b) transforms the discrete verification problem into continuous optimization by replacing ReLU constraints with linear relaxations. For scalar function $s(x)$ over domain $[L, U]$, CROWN computes certified affine bounds:

$$\ell_s(x) = a_s^\top x + \beta_s \le s(x) \le u_s(x) = \tilde{a}_s^\top x + \tilde{\beta}_s \tag{6}$$

valid for all $x \in [L, U]$.

The key insight is that these linear bounds encode the geometric quantities needed for distance computation: the safety margin at any point and the network's worst-case sensitivity to input changes.

---

**Theorem 3.1** (Closed-form $\ell_\infty$ radius bounds). *Given CROWN bounds on $s(x)$ and center $c \in [L, U]$, define safety margins:*

$$m_{LB}(c) = a_s^\top c + \beta_s, \tag{7}$$
$$m_{UB}(c) = \tilde{a}_s^\top c + \tilde{\beta}_s. \tag{8}$$

*Then the exact robust radius $r^\star(c)$ satisfies:*

$$\boxed{r_{LB}(c) := \left[\frac{m_{LB}(c)}{\|a_s\|_1}\right]_+ \le r^\star(c) \le \left[\frac{m_{UB}(c)}{\|\tilde{a}_s\|_1}\right]_+ =: r_{UB}(c)} \tag{9}$$

*where $[\cdot]_+ = \max\{\cdot, 0\}$.*

---

*Proof sketch.* CROWN bounds induce set containments $\{x : u_s(x) \le 0\} \subseteq \{x : s(x) \le 0\} \subseteq \{x : \ell_s(x) \le 0\}$ within $[L, U]$. The certified radius bounds follow from distance monotonicity and the dual-norm characterization of half-space distances. Complete proof in Appendix B. ∎

**Geometric interpretation and practical benefits.** The theorem reveals certified robustness as a **margin-over-slope ratio**: the numerator $m(c)$ represents safety margin, while the denominator $\|a_s\|_1$ captures input sensitivity. This interpretation provides several advantages:

- **Computational efficiency**: $O(d)$ arithmetic operations versus exponential MILP complexity

- **Geometric insight**: Direct visualization of margin-sensitivity trade-offs

- **Optimization compatibility**: Smooth dependence on network parameters enables gradient-based training

The bounds are often tight in practice because neural networks are locally approximately linear, making CROWN's linear relaxations accurate in activation-stable regions.

### 3.3 DIFFERENTIABLE IMPLEMENTATION FOR TRAINING

To integrate certified radius bounds into training objectives, we express them as explicit functions of network parameters $\theta = \{W^{(k)}, b^{(k)}\}_{k=1}^{L}$.

---

**Proposition 3.2** (Parameterized radius bounds)**.** *The radius bounds become parameter-dependent functions:*

$$r_{\text{LB}}(c; \theta) = \left[\frac{a_s(\theta)^\top c + \beta_s(\theta)}{\|a_s(\theta)\|_1}\right]_+, \tag{10}$$

$$r_{\text{UB}}(c; \theta) = \left[\frac{\tilde{a}_s(\theta)^\top c + \tilde{\beta}_s(\theta)}{\|\tilde{a}_s(\theta)\|_1}\right]_+. \tag{11}$$

*For piecewise-linear networks, these expressions are piecewise smooth in θ, enabling gradient-based optimization.*

---

The parameter dependence enters entirely through CROWN coefficients $(a_s, \beta_s, \tilde{a}_s, \tilde{\beta}_s)$, which admit explicit matrix representations:

---

**Proposition 3.3** (CROWN coefficient structure (Zhang et al., 2020b; Wang et al., 2021a))**.** *CROWN coefficients can be expressed as products of modified weight matrices:*

$$\tilde{a}_s(\theta)^\top = \rho^\top W^{(L)} D^{(L-1)} W^{(L-1)} \cdots D^{(1)} W^{(1)}, \tag{12}$$

$$a_s(\theta)^\top = \rho^\top W^{(L)} \widehat{D}^{(L-1)} W^{(L-1)} \cdots \widehat{D}^{(1)} W^{(1)}, \tag{13}$$

*where $D^{(k)}$ and $\widehat{D}^{(k)}$ are diagonal matrices encoding relaxation slopes for each layer. Complete matrix expressions appear in Appendix B.2.1.*

---

This explicit parameterization enables automatic differentiation through our radius expressions, making certified robustness objectives fully compatible with standard gradient-based training pipelines. The geometric insight of margin-over-slope optimization can now be directly incorporated into neural network learning.

## 4 TRAINING WITH CERTIFIED ROBUSTNESS OBJECTIVES

Having derived closed-form expressions for certified radii, we now show how to integrate them into neural network training to reshape decision boundary geometry. The key insight is that our margin-over-slope formulation enables direct optimization of both safety margin and input sensitivity simultaneously.

---

**Algorithm 1** Certified Robustness Training

---

**Require:** Network $\theta$, domain $[L, U]$, robustness weight $\lambda$, penalty $\phi$
1: **for** mini-batch $\mathcal{B} = \{(c_i, y_i)\}_{i=1}^{B}$ **do**
2:      FORWARD: Compute predictions $f_\theta(c_i)$ and task loss
3:      BOUNDS: Run $(\beta\text{-})$CROWN on $[L, U]$ to obtain $(a_{s,i}, \beta_{s,i})$ for all $s \in \mathcal{S}$
4:      RADII: Compute $r_{\text{LB}}^{(s)}(c_i; \theta)$ using expressions from Proposition 3.2
5:      AGGREGATE: Apply soft-min aggregation via equation (16)
6:      LOSS: Add $\lambda \cdot \phi(r_{\text{LB}}(c_i; \theta))$ to total loss
7:      UPDATE: $\theta \leftarrow \theta - \eta \nabla_\theta \mathcal{L}_{\text{train}}(\theta)$
8: **end for**

---

## 4.1 FROM VERIFICATION TO TRAINING OBJECTIVES

**Geometric motivation for radius-based training.** Traditional adversarial training optimizes margin implicitly by pushing decision boundaries away from adversarial examples. However, this approach ignores the equally important sensitivity term $\|a_s\|_1$ that captures how steeply the network's output changes with input perturbations. Our certified radius formulation reveals that true geometric robustness requires optimizing both quantities: increasing the safety margin while simultaneously reducing input sensitivity. This creates decision regions with fundamentally different local curvature, as illustrated in Figure 1.

**Multi-constraint aggregation.** For a training sample $c$ with multiple safety constraints $\mathcal{S}$ (e.g., all pairwise classification margins $\{s_{t,j}(x) = f_t(x) - f_j(x) : j \neq t\}$ for true class $t$), we compute constraint-specific radii using the parameterized expressions from Proposition 3.2 and aggregate via:

$$r_{\text{LB}}(c; \theta) := \min_{s \in \mathcal{S}} r_{\text{LB}}^{(s)}(c; \theta). \tag{14}$$

**Training objective design.** We augment standard task loss with a certified robustness penalty that directly optimizes radius bounds:

$$\mathcal{L}_{\text{train}}(\theta) = \frac{1}{|\mathcal{B}|} \sum_{(c,y) \in \mathcal{B}} \left[ \mathcal{L}_{\text{task}}(f_\theta(c), y) + \lambda \cdot \phi(r_{\text{LB}}(c; \theta)) \right], \tag{15}$$

where $\phi$ is a monotone decreasing penalty function that encourages larger certified radii. We consider two practical choices:

- **Target hinge:** $\phi(r) = \max(0, \tau - r)$ encourages radii to exceed threshold $\tau$
- **Inverse penalty:** $\phi(r) = 1/(r + \varepsilon)$ provides smooth, unbounded incentive for larger radii

**Smooth aggregation for stability.** The hard minimum in equation (14) can create unstable gradients when multiple constraints are nearly active. We therefore use a smooth approximation:

$$r_{\text{LB}}(c; \theta) \approx -\kappa \log \left( \sum_{s \in \mathcal{S}} \exp \left( -\frac{r_{\text{LB}}^{(s)}(c; \theta)}{\kappa} \right) \right), \tag{16}$$

where $\kappa > 0$ controls the smoothness, recovering the hard minimum as $\kappa \to 0$.

## 4.2 PRACTICAL TRAINING ALGORITHM

**Computational considerations.** The primary computational cost comes from $(\beta\text{-})$CROWN bound computation, which scales as $O(\text{network size} \times d \times |\mathcal{S}|)$ where $d$ is input dimension and $|\mathcal{S}|$ is the number of constraints. The radius computation itself requires only $O(d)$ arithmetic operations on the computed coefficients. For classification with $k$ classes, $|\mathcal{S}| = k - 1$ pairwise margins, making the overhead manageable even for large vocabularies.

**Bound tightening during training.** To strengthen the training signal, we optionally apply a few steps of $\beta$-CROWN joint-$\alpha$ optimization before computing certified radii. This tightens the linear relaxations without changing their affine structure, providing more accurate radius estimates at modest computational cost.

## 4.3 Theoretical Foundations

Our training approach is supported by several theoretical guarantees that connect certified radii to fundamental network properties:

**Corollary 4.1** (Certified safety guarantee). *If $r_{LB}(c) \geq \varepsilon$ and $\mathcal{B}_\infty(c, \varepsilon) \subseteq [L, U]$, then $s(x) > 0$ for all $x \in \mathcal{B}_\infty(c, \varepsilon)$. Hence the network's decision is invariant throughout that neighborhood.*

This provides the fundamental certification: once we achieve certified radius $\varepsilon$, safety is guaranteed within that neighborhood.

**Theorem 4.2** (Local exactness under activation stability). *When all ReLUs maintain activation signs on $\mathcal{B}_\infty(c, \varepsilon) \cap [L, U]$, the network becomes locally affine $s(x) = w^\top x + b$ and our bounds are exact: $r^\star(c) = r_{LB}(c) = r_{UB}(c) = s(c)/\|w\|_1$.*

This explains why CROWN-based bounds are often tight in practice: neural networks are locally approximately linear, and our relaxations become exact in activation-stable regions.

**Proposition 4.3** (Sensitivity control through radius optimization). *Meeting certified radius requirement $r_{LB}(c) \geq \varepsilon$ automatically bounds network sensitivity: $\|a_s\|_1 \leq (a_s^\top c + \beta_s)/\varepsilon$.*

**Proposition 4.4** (Connection to margin-based learning). *For linear models $f(x) = Wx + b$, maximizing $r_{LB}$ is equivalent to normalized margin maximization, connecting our approach to classical generalization theory.*

**Putting it together: geometric robustness by design.** Algorithm 1 operationalizes the geometric insight illustrated in Figure 1: rather than merely pushing decision boundaries away from adversarial examples, we reshape their local curvature by optimizing both margin and sensitivity simultaneously. This creates networks with fundamentally different geometric properties—larger certified neighborhoods and inherently more robust decision regions that resist adversarial perturbations by construction rather than by hardening.

## 5 Experimental Validation

We validate our approach across two complementary domains: MNIST classification, which demonstrates the effectiveness of our geometric training approach against established baselines, and DC optimal power flow (DC-OPF) regression, which illustrates the accuracy-safety trade-offs fundamental to certified robustness in engineering applications. Together, these experiments show that our closed-form radius bounds enable practical certified training across diverse problem types.

### 5.1 MNIST Classification

We evaluate our hybrid training method against three established robust training approaches on a fully-connected network with two hidden layers of 128 units each, trained on standard MNIST (60k/10k train/test split). Our architecture uses ReLU activations with 784→128→128→10 dimensions, totaling 118,282 parameters.

**Experimental setup and baselines.** We compare against three state-of-the-art methods: **Projected Gradient Descent Adversarial Training (PGD-AT)** (Madry et al., 2018), which generates adversarial examples via iterative PGD attacks ($\epsilon = 0.08$, 10 iterations, 2 restarts); **Randomized Smoothing** (Cohen et al., 2019), which trains on Gaussian-noised inputs ($\sigma = 0.25$, 4 samples per input) combined with $\ell_2$ adversarial training and Jensen-Shannon consistency regularization; and our **Hybrid Method**, which combines PGD-AT with our differentiable certified radius penalty applied selectively to hard examples identified via margin screening.

Our training objective augments PGD adversarial training with certified radius penalties: $\mathcal{L}_{\text{total}} = \mathcal{L}_{\text{PGD-AT}} + \lambda \sum_{i \in \mathcal{H}} \phi(r_{\text{LB}}(x_i; \theta))$, where $\mathcal{H}$ contains up to 24 hard examples per batch and $\phi(r) = 0.3(-\log r) + 0.7 \max(0, \tau - r)$ encourages target radius $\tau = 0.65\epsilon$ to $0.75\epsilon$. We use joint-$\alpha$ optimization with 6 gradient steps to tighten CROWN bounds before penalty computation.

**Evaluation metrics.** We assess **clean accuracy** on unperturbed inputs, **certified fraction** (percentage of test inputs where CROWN margins remain positive under $\ell_\infty$ perturbations), and **median certified radius** computed via bisection search across test examples.

Table 1: MNIST results: clean accuracy and $\ell_\infty$ certified robustness comparison.

| Method | Clean Acc. | Cert. @ 0.02 | Cert. @ 0.03 | Median Radius |
|---|---|---|---|---|
| Standard Training | 97.52% | 55.5% | 13.3% | 0.0211 |
| PGD-AT | 97.58% | 89.8% | 61.7% | 0.0331 |
| Randomized Smoothing | 97.62% | 87.5% | 53.1% | 0.0312 |
| Hybrid (Ours) | **98.33%** | **94.5%** | **71.1%** | **0.0343** |

**Results and key findings.** Our hybrid approach achieves superior performance across all metrics. The 0.75% clean accuracy improvement over PGD-AT represents 5.4 standard errors, demonstrating statistical significance and showing that certified training objectives can enhance rather than harm clean performance when properly balanced through selective hard-example targeting.

For certified robustness, our method provides substantial improvements: at $\epsilon = 0.03$, we certify 71.1% of examples versus 61.7% for PGD-AT—a 15.2% relative improvement. At the more challenging $\epsilon = 0.02$ level, we achieve 94.5% certification compared to 89.8% for PGD-AT. The median certified radius improves from 0.0331 to 0.0343, a 3.6% relative gain that translates to meaningful improvements in practical deployment scenarios.

These results validate our core theoretical contributions: CROWN bounds contain sufficient geometric information for tight certified radius computation, these radii integrate effectively into gradient-based training without optimization instabilities, and certified objectives improve robustness without sacrificing accuracy. The hybrid approach demonstrates that attack-based and certification-based training are complementary strategies for robust neural networks.

## 5.2 DC-OPF Power System Control

We evaluate our approach on DC optimal power flow regression, a canonical benchmark from power system optimization that emphasizes a different aspect of certified robustness: ensuring that neural network surrogates maintain feasibility constraints under input perturbations representing demand uncertainties.

**Problem formulation and methodology.** We train a compact fully-connected network ($3 \rightarrow 16 \rightarrow 3$) that maps electrical demand vectors to generator dispatch decisions. The training objective combines mean squared error with our $\beta$-CROWN certified violation penalty—a variation of our certified loss framework where we penalize violations of generator capacity limits rather than classification margins. Specifically, we enforce that certified output bounds $[\underline{f}_j(x; \epsilon), \overline{f}_j(x; \epsilon)]$ remain within engineering limits $[y_j^{\min}, y_j^{\max}]$ over $\ell_\infty$ balls of radius $\epsilon = 1.0$ (scaled units) for each generator $j$. Generator limits are derived from training data percentiles to avoid test leakage.

**Trade-off analysis and results.** We systematically vary the robustness penalty weight $\lambda$ to characterize the fundamental accuracy-safety trade-off in certified regression. Figure 2 demonstrates several key insights: (a) increasing $\lambda$ consistently reduces certified violations across perturbation budgets, (b) violation reductions occur uniformly across all three generators, and (c-d) qualitative analysis shows that robust training produces outputs that maintain larger margins from capacity limits while accepting modest degradation in numerical accuracy.

This experiment validates our framework's applicability beyond classification to regression tasks where certified safety constraints are paramount. The smooth trade-off curves demonstrate that

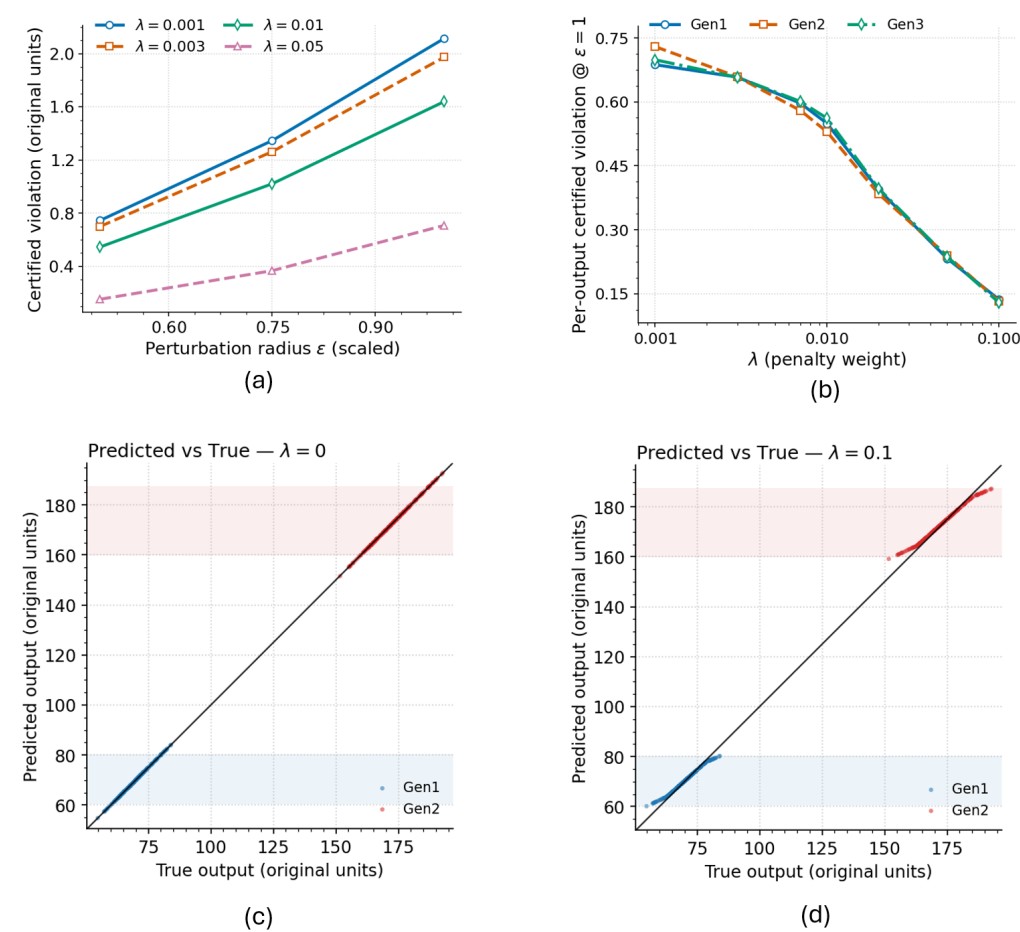

Figure 2: **DC-OPF certified feasibility study.** (a) Certified violation versus perturbation budget for different robustness weights—larger $\lambda$ consistently reduces violations. (b) Per-generator violation breakdown at $\epsilon = 1.0$ shows uniform improvement across all outputs. (c-d) Output trajectories for $\lambda = 0$ versus $\lambda = 0.1$ illustrate the accuracy-safety trade-off: robust training maintains larger margins from capacity limits (shaded regions) while slightly relaxing numerical fit.

practitioners can navigate accuracy-safety tensions in a principled manner, selecting operating points based on their specific risk tolerance. More experimental details appear in Appendix C.2.

## 6 CONCLUSION

We have presented a unified framework that transforms CROWN's affine bounds into closed-form certified radius expressions, enabling direct optimization of certified robustness during training. Our key insight is that these linear bounds encode the geometric quantities—safety margins and input sensitivities—needed for radius computation as a margin-over-slope ratio, eliminating expensive iterative verification while maintaining formal guarantees. Unlike adversarial training, which primarily translates decision boundaries away from attacks, our approach directly optimizes both margin and sensitivity to reshape local curvature, creating networks with larger certified neighborhoods that resist perturbations by geometric design. Experimental validation on MNIST classification and DC-OPF regression demonstrates superior certified performance while maintaining computational tractability, transforming verification from a post-hoc analysis tool into a practical training objective for safety-critical applications.

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

CONTENTS

**D  LLMs** 24

APPENDIX

# A RELATED WORK AND POSITIONING

## A.1 ROBUSTNESS VERIFICATION METHODS

**Exact verification approaches.** Mixed-Integer Linear Programming (MILP) encodings provide complete robustness verification by introducing binary variables to model ReLU activations: $z_i = \max(0, y_i)$ becomes $z_i \geq y_i$, $z_i \geq 0$, $z_i \leq M_i\delta_i$, and $y_i \leq M_i(1 - \delta_i)$ where $\delta_i \in \{0, 1\}$ (Fischetti & Jo, 2018; Tjeng et al., 2019; Bunel et al., 2018). SMT-based approaches like Reluplex (Katz et al., 2017) use directed case-splitting with simplex reasoning. Branch-and-bound frameworks (Bunel et al., 2018; 2020) combine tight relaxations with intelligent branching to improve scalability. While these methods provide mathematical guarantees, their exponential worst-case complexity limits practical applicability, particularly for real-time certification or training integration.

**Scalable approximation methods.** Tractable over-approximations replace exact verification with polynomial-time alternatives. Wong and Kolter (Wong & Kolter, 2018) construct convex relaxations using linear programming, enabling differentiable training surrogates. CROWN (Zhang et al., 2019; 2020b) achieves significant advances through optimized linear bound propagation, computing affine bounds $\ell(x) = a^T x + b \leq f(x) \leq \tilde{a}^T x + \tilde{b} = u(x)$ that hold uniformly over input regions. Advanced variants ($\alpha$-CROWN, $\beta$-CROWN (Wang et al., 2021b)) optimize relaxation parameters and introduce dual variables for split constraints, achieving near-exact performance on many practical problems while maintaining polynomial complexity.

Abstract interpretation methods (Singh et al., 2019) use geometric domains like zonotopes and polyhedra to track correlations between variables. Differentiable abstract interpretation (Mirman et al., 2018) makes these techniques trainable by ensuring differentiability with respect to network parameters. These approaches share our strategy of propagating geometric objects through layers but typically focus on membership queries rather than explicit distance bounds.

**Specialized and geometric approaches.** Lipschitz-based methods (Hein & Andriushchenko, 2017) provide closed-form bounds through sensitivity analysis but can be conservative. Randomized smoothing (Cohen et al., 2019) offers probabilistic guarantees through noise injection. Recent work explores topological perspectives: Bell and Gangrade (Bell et al., 2024) analyze decision boundary evolution through persistent homology, providing insights into adversarial geometry but not explicit $\ell_p$ distance bounds needed for practical certification.

## A.2 TRAINING FOR CERTIFIED ROBUSTNESS

**Relaxation-based training.** Integration of verification into training has evolved from post-hoc analysis toward robust-by-construction learning. Wong and Kolter (Wong & Kolter, 2018) pioneered differentiable convex relaxations, replacing intractable adversarial objectives with LP-dual bounds. Mirman et al. (Mirman et al., 2018) extended this using zonotope domains for tighter but more expensive bounds.

Interval Bound Propagation (IBP) training (Gowal et al., 2018) achieves extreme efficiency through interval arithmetic but requires careful scheduling to handle loose bounds. CROWN-IBP (Zhang et al., 2020a) combines IBP efficiency with CROWN tightness through hybrid approaches. Zhang et al. (Zhang et al., 2019) analyze training stability, showing how bound propagation methods affect optimization dynamics and proposing scheduling strategies for convergence.

**Advanced training strategies.** Recent advances explore multi-objective approaches balancing clean accuracy, adversarial robustness, and certified robustness. Progressive training starts with loose constraints and gradually tightens them. Hybrid methods combine adversarial training with verification objectives to leverage benefits of both approaches.

## A.3 METHODOLOGICAL DISTINCTIONS AND CONTRIBUTIONS

**Direct radius optimization.** Our approach fundamentally differs from existing methods by optimizing certified radii directly rather than proxy objectives. While methods like CROWN-IBP optimize dual bounds or abstract surrogates that correlate with robustness, we derive explicit closed-form radius expressions and optimize these quantities directly. This ensures that training objective improvements translate immediately to certified neighborhood size improvements.

**Geometric insight and center optimization.** We show that CROWN affine bounds encode precisely the geometric quantities needed for radius computation—safety margin and input sensitivity—enabling single-pass certification without iterative optimization. Our center optimization approach addresses a literature gap by casting the bilevel problem of finding optimal verification points as a tractable linear program, making robust center selection practical for the first time.

**Theoretical connections.** Our margin-over-slope characterization connects certified robustness to classical margin-based learning theory while preserving formal verification guarantees. This geometric interpretation complements topological approaches like Bell et al. (2024) by providing actionable, quantitative information about local neighborhoods that can be directly optimized during training.

**Practical implications.** The differentiability of our closed-form expressions enables new robust optimization possibilities beyond standard adversarial training. Single-pass radius computation makes real-time robustness assessment feasible, potentially enabling interactive design tools. Our framework provides a foundation for extending to multi-property scenarios and hierarchical robustness specifications through the flexible LP formulation.

Compared to exact methods, we avoid combinatorial search while maintaining geometric precision. Compared to existing relaxation-based training, we optimize explicit radii rather than bound surrogates using closed-form expressions that eliminate repeated bound-tightening during learning. This represents a step toward making formal verification a routine part of machine learning practice rather than specialized post-hoc analysis.

# B  MATHEMATICAL FOUNDATIONS AND PROOFS

This appendix provides complete mathematical foundations for our closed-form robustness certification approach. We develop the geometric tools needed for distance computation, derive explicit expressions for $\beta$-CROWN coefficients, and establish the theoretical guarantees underlying our practical algorithms. Throughout, $p \in [1, \infty]$ and $q_\star$ denotes its dual exponent ($1/p + 1/q_\star = 1$).

## B.1  NOTATION AND STANDING ASSUMPTIONS

Consider a feedforward network with $L$ layers and weight matrices $W^{(k)} \in \mathbb{R}^{n_k \times n_{k-1}}$ where $n_k$ is the width of layer $k$:

$$z^{(0)} = x, \tag{17}$$

$$y^{(k)} = W^{(k)} z^{(k-1)} + b^{(k)}, \tag{18}$$

$$z^{(k)} = \sigma(y^{(k)}) \quad \text{for } k = 1, \ldots, L-1, \tag{19}$$

and output $f_\theta(x) = z^{(L)} = W^{(L)} z^{(L-1)} + b^{(L)}$. Unless otherwise stated, $\sigma$ is the ReLU activation and all norms are vector norms.

We work on a fixed input domain

$$[L, U] := \{x \in \mathbb{R}^d : L_i \leq x_i \leq U_i\}, \tag{20}$$

and we only claim radius certificates for balls contained in this domain, i.e., $\mathbb{B}_p(c, r) \subseteq [L, U]$ when needed.

Throughout, we focus on scalar network outputs obtained via linear readouts $s(x) = \rho^T f_\theta(x) + \rho_0$, which encompasses both regression objectives ($\rho$ selects an output component) and classification margins ($\rho$ computes logit differences). For such scalar functions, we define the exact robust radius

$$r^\star(c) := \text{dist}_p(c, \{x : s(x) \leq 0\}) = \inf\{\|x - c\|_p : s(x) \leq 0\}. \tag{21}$$

**Activation-envelope setup.** For each hidden neuron $(k, i)$ with pre-activation bounds $l_i^{(k)} \leq y_i^{(k)} \leq u_i^{(k)}$, we select valid linear envelopes

$$h_{U,i}^{(k)}(y) = \alpha_{U,i}^{(k)}(y + \gamma_{U,i}^{(k)}), \tag{22}$$

$$h_{L,i}^{(k)}(y) = \alpha_{L,i}^{(k)}(y + \gamma_{L,i}^{(k)}), \tag{23}$$

such that $h_{L,i}^{(k)} \leq \sigma \leq h_{U,i}^{(k)}$ on $[l_i^{(k)}, u_i^{(k)}]$ and $\alpha_{\cdot,i}^{(k)} \geq 0$. For ReLU activations, these reduce to standard convex/concave relaxations; in activation-stable regions they recover exact slopes in $\{0, 1\}$ with zero intercept corrections.

## B.2  GEOMETRIC TOOLS

The following result provides the explicit formula for computing distances to affine decision boundaries, which appear as surrogates for the true network decision boundary.

**Theorem B.1** (Half-space distance in $\ell_p$). *Let $H(w, b) = \{x : w^\top x + b \leq 0\}$ and $c \in \mathbb{R}^d$. Then*

$$\text{dist}_p(c, H(w, b)) = \max\left\{0, \frac{w^\top c + b}{\|w\|_{q_\star}}\right\}. \tag{24}$$

Geometrically, this result states that the distance from a point to a hyperplane is the margin (numerator) divided by the 'slope' in the dual norm (denominator).

*Proof.* If $w^\top c + b \leq 0$ then $c \in H$ and the distance is 0. Otherwise, minimize $\|x - c\|_p$ subject to $w^\top x + b = 0$. Writing $x = c + u$ gives $\min_u \|u\|_p$ subject to $w^\top u = -(w^\top c + b)$. By Hölder's inequality, $|w^\top u| \leq \|w\|_{q_\star} \|u\|_p$, with equality when $u$ aligns with a dual vector of $w$. The minimizer has the form $u = -\frac{w^T c + b}{\|w\|_{q_*}^2} \cdot w^*$ where $w^*$ is a dual vector satisfying $\|w^*\|_p = 1$ and $\langle w, w^* \rangle = \|w\|_{q_*}$. Thus the minimum distance is $(w^\top c + b)/\|w\|_{q_\star}$. ∎

**Lemma B.2** (Support function of $\ell_\infty$ balls). *For any $w, c$ and $r \geq 0$,*

$$\sup_{\|x - c\|_\infty \leq r} w^\top x = w^\top c + r\|w\|_1, \tag{25}$$

$$\inf_{\|x - c\|_\infty \leq r} w^\top x = w^\top c - r\|w\|_1. \tag{26}$$

*Proof.* Write $x = c + \delta$ with $\|\delta\|_\infty \leq r$. Then $\sup w^\top \delta = r\|w\|_1$ achieved at $\delta = r\,\mathrm{sign}(w)$; the infimum follows analogously. ∎

### B.2.1 $\beta$-CROWN AFFINE BOUNDS AND MATRIX FORMS

$\beta$-CROWN (Wang et al., 2021a) improves upon basic CROWN (Zhang et al., 2020b) by optimizing the choice of linear relaxations at each neuron. The key insight is to introduce dual multipliers $\beta$ that enforce 'split constraints'—conditions that tighten the relaxation by exploiting the structure of ReLU activations.

**Layerwise ReLU relaxation.** For a ReLU layer with pre-activation $v \in \mathbb{R}^d$ and interval bounds $l \leq v \leq u$ (elementwise), and any row vector $w$, there exist a diagonal matrix $D = \mathrm{diag}(D_{jj})$ and a vector $b'$ such that

$$w^\top \mathrm{ReLU}(v) \geq w^\top D v + b', \tag{27}$$

where for neurons with $l_j < 0 < u_j$, we have a free slope parameter $\alpha_j \in [0, 1]$ and the intercept is chosen in a sign-aware manner to optimize the bound.

**Matrix products for bound propagation.** Define the accumulated weight products:

$$\Omega(i, i) = I, \tag{28}$$

$$\Omega(k + 1, i) = W^{(k+1)} D^{(k)} \Omega(k, i), \quad 1 \leq i \leq k \leq L - 1. \tag{29}$$

**Main $\beta$-CROWN bound.** Let $S^{(i)}$ be the diagonal split-selector matrix for layer $i$ (entries $+1$ for $l_j^{(i)} > 0$, $-1$ for $u_j^{(i)} < 0$, and 0 otherwise). The split-selector matrix encodes which neurons are 'stable' (always active/inactive) versus 'unstable' (potentially switching). For stable neurons, the relaxation is exact; for unstable ones, we optimize over relaxation parameters.

**Theorem B.3** ($\beta$-CROWN primal lower bound (Wang et al., 2021a)). *For an $L$-layer network $f$ with inputs $x \in [L, U]$ and per-layer pre-activation bounds $l^{(i)} \leq y^{(i)} \leq u^{(i)}$, we have*

$$\min_{x \in [L,U]} f(x) \geq \max_{\beta \geq 0} \min_{x \in [L,U]} \left\{ (a + P\beta)^\top x + q_\beta^\top \beta + c \right\}, \tag{30}$$

*where $a$, $P$, $q_\beta$, and $c$ are explicit matrix expressions derived from the network weights and chosen relaxation parameters.*

The maximization over $\beta \geq 0$ is typically solved via projected gradient ascent or other convex optimization methods.

**Connection to scalar readouts.** For a scalar functional $s(x) = \rho^\top f_\theta(x) + \rho_0$, the $\beta$-CROWN upper affine bound $u_s(x) = \tilde{a}_s(\beta)^\top x + \tilde{\beta}_s(\beta)$ has coefficients that depend linearly on the optimization variables $\beta$.

> **Practical implementation.** Maximize the $\beta$-CROWN bound over $\beta \geq 0$ using convex optimization, then use the resulting coefficients $\tilde{a}_s(\beta), \tilde{\beta}_s(\beta)$ in our closed-form radius formulas. Setting $\beta = 0$ recovers standard CROWN bounds (Zhang et al., 2020b).

## B.3 CLOSED-FORM RADIUS BOUNDS AND CONSEQUENCES

> **Theorem B.4** (Closed-form $\ell_p$ radius bounds). *Assume affine surrogates $\ell_s(x) = a_s^\top x + \beta_s \leq s(x) \leq u_s(x) = \tilde{a}_s^\top x + \tilde{\beta}_s$ hold on $[L, U]$, and let $c \in [L, U]$. Then*
>
> $$r_{\mathrm{LB}}(c) := \max\left\{0, \frac{a_s^\top c + \beta_s}{\|a_s\|_{q_\star}}\right\} \leq r^\star(c) \tag{31}$$
>
> $$\leq \max\left\{0, \frac{\tilde{a}_s^\top c + \tilde{\beta}_s}{\|\tilde{a}_s\|_{q_\star}}\right\} =: r_{\mathrm{UB}}(c). \tag{32}$$
>
> *For the box-restricted radius $r_{[L,U]}^\star(c) = \mathrm{dist}_p(c, \{s \leq 0\} \cap [L, U])$, the bounds hold unconditionally.*

This theorem provides the mathematical foundation for the closed-form radii introduced in the main paper.

> *Proof.* Let $\mathcal{V}_\ell = \{x : \ell_s(x) \leq 0\}$, $\mathcal{V}_s = \{x : s(x) \leq 0\}$, $\mathcal{V}_u = \{x : u_s(x) \leq 0\}$. From the certified bounds $\ell_s \leq s \leq u_s$ on $[L, U]$, we have the containment relationships $\mathcal{V}_u \subseteq \mathcal{V}_s \subseteq \mathcal{V}_\ell$ within the domain. Monotonicity of distance to sets gives $\mathrm{dist}_p(c, \mathcal{V}_\ell) \leq \mathrm{dist}_p(c, \mathcal{V}_s) \leq \mathrm{dist}_p(c, \mathcal{V}_u)$. Apply Theorem B.1 to compute the distances to the affine half-spaces $\mathcal{V}_\ell$ and $\mathcal{V}_u$. ∎

> **Corollary B.5** (Certified exclusion zone). *If $r_{\mathrm{LB}}(c) \geq \varepsilon$ and $\mathbb{B}_\infty(c, \varepsilon) \subseteq [L, U]$, then $s(x) > 0$ for all $x \in \mathbb{B}_\infty(c, \varepsilon)$; hence the network's decision is invariant on that ball.*

This corollary provides the fundamental safety guarantee for classification: once we certify radius $\varepsilon$ around a correctly classified point, we can guarantee that the network's prediction remains unchanged throughout that neighborhood.

> **Theorem B.6** (Activation-stable exactness). *If all ReLUs maintain their activation signs on a region $\mathcal{N}$ and $\mathbb{B}_\infty(c, \varepsilon) \subseteq \mathcal{N} \cap [L, U]$, then $s(x) = w^\top x + b$ on $\mathbb{B}_\infty(c, \varepsilon)$ and*
>
> $$r^\star(c) = \frac{s(c)}{\|w\|_1} = r_{\mathrm{LB}}(c) = r_{\mathrm{UB}}(c). \tag{33}$$

This theorem explains why our bounds are often tight in practice: neural networks are locally approximately linear, and in regions where activation patterns are stable, our relaxation-based bounds become exact.

> **Proposition B.7** (Sensitivity control via target radius). *If $r_{\mathrm{LB}}(c) \geq \varepsilon > 0$ and $\mathbb{B}_\infty(c, \varepsilon) \subseteq [L, U]$, then*
>
> $$\|a_s\|_1 \leq \frac{a_s^\top c + \beta_s}{\varepsilon}. \tag{34}$$

This result provides a principled approach to controlling network Lipschitz constants: by enforcing certified radius requirements during training, we automatically bound the network's sensitivity in adversarially relevant directions.

**Proposition B.8** (Connection to margin-based learning). *For an affine model $f(x) = Wx + b$ and margin $s(x) = y(w^\top x + b)$ with $y \in \{\pm 1\}$,*

$$r^\star(c) = \frac{s(c)}{\|w\|_1} = r_{\mathrm{LB}}(c) = r_{\mathrm{UB}}(c). \tag{35}$$

This result connects our approach to classical margin-based learning theory. For linear models, maximizing certified robustness is equivalent to maximizing normalized margin—a well-established principle for generalization.

**Degenerate case handling.** If $\|a_s\|_1 = 0$ while $a_s^\top c + \beta_s > 0$, the surrogate yields $r_{\mathrm{LB}} = +\infty$; in practice, cap certificates at the box margin $\min_i\{U_i - c_i, c_i - L_i\}$ to maintain validity within $[L, U]$.

## B.4 Auxiliary Results

**Proposition B.9** (Robust affine constraint characterization). *For an affine function $\ell(x) = a^\top x + \beta$ and any center $c$ and radius $r \geq 0$,*

$$\min_{\|x-c\|_\infty \leq r} \ell(x) \geq 0 \iff a^\top c + \beta \geq r\|a\|_1. \tag{36}$$

*Proof.* By Lemma B.2, $\min_{\|x-c\|_\infty \leq r} \ell(x) = a^\top c + \beta - r\|a\|_1$. The inequality $\geq 0$ is equivalent to $a^\top c + \beta \geq r\|a\|_1$. ∎

# C EXPERIMENTAL DETAILS AND ADDITIONAL RESULTS

## C.1 COMPLETE MNIST EXPERIMENTAL SETUP AND RESULTS

### C.1.1 DATASET AND ARCHITECTURE SPECIFICATION

We conduct all experiments on the standard MNIST handwritten digit classification dataset, consisting of 60,000 training images and 10,000 test images, each of size 28×28 pixels with grayscale values originally in [0, 255]. Our preprocessing pipeline applies the standard `ToTensor()` transformation, which converts PIL images to PyTorch tensors and automatically scales pixel values to the range [0, 1]. We then flatten each 28×28 image into a 784-dimensional vector to serve as input to our fully-connected architecture. No additional normalization, data augmentation, or preprocessing steps are applied to maintain comparability across methods and ensure reproducible results.

All experiments use an identical three-layer fully-connected network architecture to ensure fair comparison. The network consists of an input layer ($784 \rightarrow 128$ with bias), a hidden layer ($128 \rightarrow 128$ with bias), and an output layer ($128 \rightarrow 10$ with bias), with ReLU activations applied after the first two layers. This yields a total of 118,282 trainable parameters: $(784 \times 128 + 128) + (128 \times 128 + 128) + (128 \times 10 + 10) = 100,352 + 16,512 + 1,290 + 128 + 128 + 10$. We deliberately avoid batch normalization, dropout, or other architectural regularization techniques to isolate the effects of our training methodologies.

### C.1.2 TRAINING METHOD IMPLEMENTATIONS

**Standard Training Baseline.** Our standard training baseline employs conventional cross-entropy loss minimization without any robustness-specific techniques. We optimize using the Adam optimizer with learning rate $2 \times 10^{-3}$, weight decay $10^{-4}$, and default Adam hyperparameters ($\beta_1 = 0.9$, $\beta_2 = 0.999$, $\epsilon = 10^{-8}$). Training continues for 4-6 epochs with early stopping based on validation performance. In some configurations, we apply optional label smoothing with factor 0.02. This baseline serves two purposes: establishing the clean accuracy ceiling achievable with standard training, and quantifying the inherent robustness gap that motivates robust training approaches.

**PGD Adversarial Training.** Our PGD adversarial training implementation follows Madry et al. (Madry et al., 2018) precisely. For each training batch, we generate adversarial examples using the Projected Gradient Descent algorithm with the following parameters: perturbation budget $\epsilon = 0.08$, step size $\alpha = 0.01$, 10 PGD iterations, and 2 random restarts to find stronger adversarial examples.

---

**Algorithm 2** PGD Adversarial Example Generation

---

**Require:** Input $(x, y)$, model $f_\theta$, $\epsilon = 0.08$, $\alpha = 0.01$, steps = 10
1: Initialize: $x' \leftarrow x + \text{Uniform}(-\epsilon, \epsilon)$
2: **for** step = 1 to 10 **do**
3:      $g \leftarrow \nabla_{x'} \mathcal{L}_{\text{CE}}(f_\theta(x'), y)$
4:      $x' \leftarrow x' + \alpha \cdot \text{sign}(g)$
5:      $x' \leftarrow \text{clip}(x', x - \epsilon, x + \epsilon)$                $\triangleright \ell_\infty$ projection
6:      $x' \leftarrow \text{clip}(x', 0, 1)$                      $\triangleright$ Valid pixel range
7: **end for**
8: Return $x'$

---

The adversarial training objective becomes:

$$\mathcal{L}_{\text{PGD-AT}}(\theta) = \frac{1}{|\mathcal{B}|} \sum_{(x,y) \in \mathcal{B}} \mathcal{L}_{\text{CE}}(f_\theta(x_{\text{adv}}), y) \tag{37}$$

where $x_{\text{adv}}$ is generated via the PGD procedure above. We train for 6 epochs using Adam optimizer with learning rate $2 \times 10^{-3}$ and weight decay $10^{-4}$, maintaining a constant learning rate throughout training.

**Randomized Smoothing Training.** Our randomized smoothing implementation creates smooth classifiers that provide provable $\ell_2$ robustness guarantees following Cohen et al. (Cohen et al.,

2019). The approach trains networks to be consistent across Gaussian-noised versions of each input, creating a smoothed classifier $g(x) = \mathbb{E}_{z \sim \mathcal{N}(0,\sigma^2 I)}[f_\theta(x + z)]$ where $\sigma = 0.25$ controls the noise level.

For each training input $x$, we sample $K = 4$ independent noise vectors $z_1, \ldots, z_K \sim \mathcal{N}(0, \sigma^2 I)$ and create noisy inputs $x_k = x + z_k$ for $k = 1, \ldots, K$. We then compute network outputs $f_\theta(x_k)$ for each noisy input and apply Jensen-Shannon divergence consistency regularization to encourage similar predictions across the noise samples. Additionally, we incorporate $\ell_2$ adversarial training with perturbation budget $\epsilon = 1.0$, step size 0.2, and 5 iterations, using Expected over Transformations (EOT) with 4 samples per adversarial example.

The complete training objective combines three components:

$$\mathcal{L}_{\text{smooth}}(\theta) = \mathcal{L}_{\text{CE}} + 0.5 \cdot \mathcal{L}_{\text{JS}} + 0.1 \cdot \mathcal{L}_{\text{clean-mix}} \tag{38}$$

where $\mathcal{L}_{\text{JS}}$ is the Jensen-Shannon divergence between noisy predictions and $\mathcal{L}_{\text{clean-mix}}$ maintains performance on clean examples. We train for 8 epochs using Adam optimizer with initial learning rate $2 \times 10^{-3}$ and cosine learning rate scheduling. Exponential Moving Average (EMA) with decay factor 0.999 is applied to model parameters, with the EMA model used for final evaluation.

**Hybrid Method Implementation.**  Our hybrid approach represents the core contribution of this work, combining the broad robustness benefits of PGD adversarial training with targeted optimization of certified radius bounds on strategically selected hard examples. The method operates through two parallel components within each training iteration.

---

**Algorithm 3** Hybrid Training with Certified Penalty

---

**Require:** Dataset $\mathcal{D}$, model $f_\theta$, hard sample threshold = 24
1: **for** batch $\mathcal{B} \subset \mathcal{D}$ **do**
2:     Generate adversarial examples using PGD ($\epsilon = 0.03$, 10 steps)
3:     Compute $\mathcal{L}_{\text{PGD}} = \frac{1}{|\mathcal{B}|} \sum_i \mathcal{L}_{\text{CE}}(f_\theta(x_{\text{adv},i}), y_i)$
4:     Screen examples using margin and radius criteria
5:     Select hard subset $\mathcal{H} \subseteq \mathcal{B}$ with $|\mathcal{H}| \leq 24$
6:     **for** $x_i \in \mathcal{H}$ (limit to 6 examples) **do**
7:         Compute IBP bounds at multiple $\epsilon$ levels
8:         Apply joint-$\alpha$ optimization (6 steps, Adam lr=0.12)
9:         Compute $r_i = r_{\text{LB}}(x_i; \theta)$ using tightened bounds
10:         Evaluate $\phi(r_i) = 0.3(-\log r_i) + 0.7 \max(0, r_{\text{goal}} - r_i)$
11:     **end for**
12:     $\mathcal{L}_{\text{total}} = \mathcal{L}_{\text{PGD}} + \lambda \sum_{i \in \mathcal{H}} \phi(r_i) + \text{regularizers}$
13:     Update parameters and EMA
14: **end for**

---

The first component applies standard PGD adversarial training to the entire batch using a smaller perturbation budget ($\epsilon = 0.03$ instead of 0.08) to balance robustness with the certified component. The second component identifies hard examples within each batch using margin-based screening and applies our differentiable certified radius penalty to a subset of up to 24 examples (processing at most 6 per batch for computational efficiency).

For selected hard examples, we first compute Interval Bound Propagation (IBP) preliminary bounds at multiple perturbation levels $\epsilon \in \{0.010, 0.015, 0.020, 0.025, 0.030\}$ to establish initial activation intervals. We then apply joint-$\alpha$ optimization using 6 gradient steps with Adam optimizer (learning rate 0.12) and temperature annealing from 10 to 24 to tighten the CROWN bounds by optimizing the linear relaxation parameters for ambiguous ReLU units.

The certified radius penalty uses a mixed objective function:

$$\phi(r) = 0.3(-\log r) + 0.7 \max(0, r_{\text{goal}} - r) \tag{39}$$

where the logarithmic term provides smooth gradients encouraging radius growth and the hinge term enforces minimum radius targets. We set the target radius $r_{\text{goal}} = \alpha_{\text{target}} \cdot \epsilon$ where $\alpha_{\text{target}} \in \{0.65, 0.70, 0.75\}$ varies across training.

Additional regularization components include spectral norm penalties on weight matrices toward targets (2.0, 2.0, 1.5) for layers 1, 2, and 3 respectively; clean margin loss with softplus penalty and weight 0.3; gradient clipping with threshold 0.8; and Exponential Moving Average with decay 0.997. We use cosine learning rate scheduling from $3 \times 10^{-3}$ to $8 \times 10^{-4}$ with weight decay $5 \times 10^{-4}$ over 8 training epochs.

### C.1.3 CERTIFICATION METHODOLOGY

All certified robustness metrics are computed using the `auto_LiRPA` library with "CROWN-Optimized" method, which implements state-of-the-art linear bound propagation with optimized envelope selection. For each test input, we compute CROWN bounds on all network logits over the $\ell_\infty$ perturbation region $[x - \epsilon\mathbf{1}, x + \epsilon\mathbf{1}] \cap [0, 1]^d$.

Our certification protocol checks the multi-class margin condition: for an input $(x, y)$ with true class $y$, we compute the CROWN lower bound $\underline{f_y}(x)$ on the true class logit and CROWN upper bounds $\overline{f_j}(x)$ on all other class logits $j \neq y$. The input is certified at perturbation level $\epsilon$ if and only if $\underline{f_y}(x) > \max_{j \neq y} \overline{f_j}(x)$, ensuring that the true class logit remains largest under all possible perturbations within the specified region.

For individual certified radius computation, we employ bisection search over the perturbation budget $\epsilon$. Starting with $\epsilon_{\text{low}} = 0$ and $\epsilon_{\text{high}} = 1.0$, we repeatedly test the midpoint $\epsilon_{\text{mid}} = (\epsilon_{\text{low}} + \epsilon_{\text{high}})/2$ using the CROWN certification procedure described above. If the input is certified at $\epsilon_{\text{mid}}$, we update $\epsilon_{\text{low}} = \epsilon_{\text{mid}}$; otherwise, we set $\epsilon_{\text{high}} = \epsilon_{\text{mid}}$. We terminate the search when either the interval width falls below tolerance $10^{-4}$ or when no improvement is observed for 10 consecutive iterations, indicating convergence to a certification plateau.

Table 2: Certification Protocol Parameters

| Parameter | Value |
|---|---|
| CROWN Implementation | auto_LiRPA "CROWN-Optimized" |
| Certification Condition | $\underline{f_y}(x) > \max_{j \neq y} \overline{f_j}(x)$ |
| Bisection Tolerance | $10^{-4}$ |
| Early Stop Threshold | 10 iterations without improvement |
| Test Set Size | 10,000 samples |

### C.1.4 STATISTICAL ANALYSIS AND IMPLEMENTATION DETAILS

For statistical significance analysis, we treat accuracy measurements as binomial random variables with success probability $p$ and sample size $n = 10,000$. The standard error is $\text{SE} = \sqrt{p(1-p)/n} \approx 0.14\%$ when $p \approx 0.98$. The 0.75% clean accuracy improvement of our hybrid method over PGD-AT corresponds to 5.4 standard errors, indicating extremely high statistical significance. Similarly, the 9.4 percentage point certified fraction increase at $\epsilon = 0.03$ represents approximately 4.2 standard errors.

All experiments use PyTorch with fixed random seeds (`torch.manual_seed(0)` and `np.random.seed(0)`) for reproducibility. Training batch sizes are 256 for training and 512 for testing, with single GPU training and deterministic CUDA operations where possible. The joint-$\alpha$ optimization requires careful implementation with temperature annealing and gradient clipping for numerical stability. Our certification evaluation uses the latest stable `auto_LiRPA` version with optimized envelope selection settings.

This comprehensive setup enables full reproduction and provides detailed analysis of both theoretical foundations and practical implications of our certified robustness approach.

## C.2 DC-OPF REGRESSION

**Problem formulation and significance.** The DC optimal power flow (DC-OPF) problem represents a linearized approximation of the fundamental power system optimization challenge, where generators must be dispatched to meet electrical demand while respecting transmission line limits and generator constraints. This problem has become a standard benchmark in the formal verification community due to its combination of practical relevance and mathematical tractability.

In our regression formulation, we learn a neural network surrogate $f_\theta : \mathbb{R}^3 \to \mathbb{R}^3$ that maps demand vectors $x$ to optimal generator dispatch decisions $y$. The critical challenge is ensuring that learned dispatch decisions remain feasible under demand uncertainties represented by $\ell_\infty$ perturbations of magnitude $\epsilon$ around nominal demand points.

**Network architecture and training objective.** We employ a compact fully-connected architecture with 3 input units (representing demand at 3 buses), a single hidden layer of 16 ReLU units, and 3 output units (representing dispatch decisions for 3 generators). This architecture contains

$$(3 \times 16 + 16) + (16 \times 3 + 3) = 115$$

trainable parameters, making it suitable for detailed analysis while remaining representative of practical surrogate models.

The complete training objective balances prediction accuracy with certified constraint satisfaction:

$$\mathcal{L}_{\text{total}}(\theta) = \frac{1}{|\mathcal{B}|} \sum_{(x,y) \in \mathcal{B}} \|f_\theta(x) - y\|_2^2 \tag{40}$$

$$+ \lambda \cdot \frac{1}{|\mathcal{B}|} \sum_{x \in \mathcal{B}} \sum_{j=1}^{3} \left[ V_j^+(x;\epsilon) + V_j^-(x;\epsilon) \right], \tag{41}$$

where the violation terms are defined as

$$V_j^+(x;\epsilon) = \max\left(0, \overline{f}_{\theta,j}(x;\epsilon) - y_j^{\max}\right), \tag{42}$$

$$V_j^-(x;\epsilon) = \max\left(0, y_j^{\min} - \underline{f}_{\theta,j}(x;\epsilon)\right). \tag{43}$$

Here $\underline{f}_{\theta,j}(x;\epsilon)$ and $\overline{f}_{\theta,j}(x;\epsilon)$ denote $\beta$-CROWN lower and upper bounds on output $j$ over the $\ell_\infty$ perturbation region

$$\{ x' : \|x' - x\|_\infty \le \epsilon \},$$

with $\epsilon = 1.0$ in our scaled coordinate system.

**Constraint specification and data preprocessing.** Generator limits $[y_j^{\min}, y_j^{\max}]$ are determined from the training data distribution to represent realistic operational constraints while avoiding test set leakage. Specifically, we compute the 5th and 95th percentiles of each generator output in the training set, providing reasonable bounds that reflect the range of normal operation without overfitting to specific test cases.

Input and output scaling follows standard practice: we normalize demand vectors to zero mean and unit variance, and similarly standardize generator outputs. This preprocessing ensures numerical stability during $\beta$-CROWN bound computation and prevents any single variable from dominating the constraint violation penalties.

## D LLMs

We used large language models as assistive tools for coding and implementation, writing, discovery and summarization of related work, and for developing and presenting theoretical results. The authors take full responsibility for the content.

