# OpenReview forum: "Certified Robustness Training: Closed-Form Certificates via CROWN"
_ICLR.cc/2026/Conference — Submitted to ICLR 2026_

### Official Review · Reviewer_F3jr · 2025-10-24

**Soundness:** 3
**Presentation:** 2
**Contribution:** 1
**Rating:** 0
**Confidence:** 5

**Summary:**

This paper utilizes the CROWN framework to acquire estimates of classification margins for ReLU neural networks. Subsequently, by penalizing the training loss, they seek to increase the margins of classification, thereby increasing the robustness of the classifiers and defending against adversarial attacks. Experimental results are provided on the MNIST dataset and a power system control.

**Strengths:**

Unfortunately, nothing specific caught my attention to be strength-worthy.

**Weaknesses:**

This paper does not rise to the level of an ML conference, yet ICLR. It is more fitting for a course project.

The novelty in this paper is indeed very limited and the authors do not properly establish baselines or compare with them. There are works that are indeed close to this that are not properly cited or compared with.

The paper is repetitive and this raises concerns regarding the potential significant use of LLMs in writing the paper.

The authors make some geometrical claims over and over (in line with the repetitiveness) without really providing any theoretical, empirical, or even geometrical justifications.

The methodology of the paper is in essence similar to [1] and [2], neither of which have been cited or compared with. More specifically, [1] has performed the same line of thought and reasoning, but instead of using CROWN, they utilize the Lipschitz constant to provide bounds. Indeed, CROWN can potentially provide tighter bounds, but it is important to give credit to the original work. I am aware that [1] studies $ l_2 $ bounds instead of the $ l_\infty $ done in here, but nonetheless, the core of the methods is the same.

There are many baselines that are missing in the experiment section. Comparison has to be made against many adversarial training methods like [2], TRADES, etc.

The experiments are very limited and very small-scale, not meeting the standards of ICLR. There is only an experiment on the MNIST dataset, which is a very simple task and pretty saturated, and then on a power system setup, which is even smaller. These are not acceptable.
Moreover, the power system experiment lacks any comparison with other work.


Some other comments:
- I understand the appeal of the differentiability of the CROWN bounds. However, the authors are, in a way, selling this too much, given that they are using ReLU networks, which are technically not differentiable!
- 33 (and many other places): The authors claim that adversarial training just moves the decision boundaries without changing geometry. What is the proof for this? How can you claim that this is indeed what is happening.
- fig 1: the boundaries are all the same in the left and right. what has changed between adversarial training vs your training? No margin has changed.
- 81-82: 'this transforms expensive post-hoc verification into an ...' The expensive part of the verification is due to the branching, which you are not doing, right? If so, you haven't really done anything and haven't added any contribution and are just doing bare CROWN. So, why are you saying that this has transformed ...?
- 144: Can you use any scalar function s(x)?? Doesn't CROWN only work with linear functions of the output of the network? Can you for example, handle the log-sum-exp on the output of the network?

CROWN provides linear upper and lower bounds. How is this showing the local geometric structure that you keep claiming it does? Between these two hyperplanes that CROWN finds, the geometric structure might as well be any complex non-linear non-convex function.




[1] Fazlyab M, Entesari T, Roy A, Chellappa R. Certified robustness via dynamic margin maximization and improved lipschitz regularization. Advances in Neural Information Processing Systems. 2023 Dec 15;36:34451-64.

[2] Xu Y, Sun Y, Goldblum M, Goldstein T, Huang F. Exploring and exploiting decision boundary dynamics for adversarial robustness. arXiv preprint arXiv:2302.03015. 2023 Feb 6.

**Questions:**

See Weaknesses.

---

> ### Author Response · Authors · 2025-12-03
>
> We thank the reviewer for their rich and detailed feedback.
>
> ### Weakness 1 – Novelty and relation to prior margin / geometry works ([1], [2])
>
> - Reviewer considers the **novelty very limited** and says our method is essentially similar to:
>   - [1] Fazlyab et al., “Certified Robustness via Dynamic Margin Maximization and Improved Lipschitz Regularization” (dynamic margin + Lipschitz regularization), and
>   - [2] Xu et al., “Exploring and Exploiting Decision Boundary Dynamics for Adversarial Robustness” (DyART, margin dynamics).
> - They note that these works are **not cited or compared** in the current draft, and argue that the “core of the methods is the same”.
>
> We thank the reviewer for pointing out these important connections. We agree that our current related-work discussion underplays the relationship to [1] and [2], and we will correct this in the revision.
>
> 1. **Add explicit comparisons and citations.**
>    In the related-work section we will add a dedicated paragraph “Margin-based and Lipschitz-based robust training” that:
>    - summarizes [1] as a method that **maximizes margins while regularizing a Lipschitz upper bound**, using differentiable Lipschitz estimates to obtain *global* certified bounds, and
>    - summarizes [2] as a study of **decision-boundary dynamics under adversarial training**, which derives gradients of *attack-based margins* and proposes DyART to prioritize small margins.
>    We will clearly cite these works and acknowledge that they share the high-level goal of **geometry-aware robust training**.
>
> 2. **Clarify what is new in our approach.**
>    We will then contrast these works with our contribution, emphasizing that:
>    - We work with **CROWN linear relaxations** to derive a **closed-form *certified* radius lower bound** based on a margin-over-slope ratio. This radius is guaranteed with respect to the same linear bounds used in verification, not just a heuristic or global Lipschitz bound.
>    - Our objective is **local and certificate-driven**: for each point we optimize a radius derived from half-space geometry of CROWN bounds, rather than a global Lipschitz constant [1] or attack-based margin dynamics [2].
>    - We show that this closed-form certified radius can be **differentiated and optimized directly**, and we integrate it into a **hybrid verification-guided training scheme** that also covers a regression task, which [1] and [2] do not address.
>
> 3. **Rephrase contribution claims accordingly.**
>    We will adjust the introduction and contribution bullets to state that our main novelty is:
>    - turning CROWN’s affine bounds into a **closed-form differentiable certified radius** and
>    - demonstrating a **proof-of-concept** hybrid training procedure based on this radius,
>    rather than claiming to be the first to connect robustness training and geometric margins.

---

> ### Author Response · Authors · 2025-12-03
>
> ### Weakness 2 – Missing baselines and limited experimental scale
>
> - Reviewer states that many **important baselines are missing**, including:
>   - other adversarial training methods such as TRADES and the margin-based DyART method [2],
>   - certified training schemes such as [1], and
>   - any comparison for the regression experiment.
> - They also emphasize that our **MNIST network and radii are very small** compared to recent certified-training papers, and conclude that the current experiments are more like a course project than an ICLR paper.
>
> We agree that the current experimental section is limited and we appreciate the pointers to additional baselines.
>
> 1. **Clarify the intended empirical scope.**
>    We will explicitly state in Section 4 that our experiments are meant as **proof-of-concept demonstrations** of the new geometric objective, not as fully scaled comparisons to the most recent and strongest defenses. We will soften any wording that could be interpreted as claiming state-of-the-art robustness and emphasize that our contribution is conceptual / methodological.
>
> 2. **Add stronger baselines on MNIST where feasible.**
>    Within our compute and page budget, we plan to augment the MNIST experiments as follows:
>    - Add a **CROWN-based certified loss baseline**, where we train on a standard adversarial cross-entropy upper bound computed from the same CROWN bounds, to isolate the effect of the margin-over-slope objective versus using the bounds only in the loss.
>    - Add at least one additional **adversarial-training baseline**, such as TRADES, implemented on the same architecture and evaluated at the same epsilon and norm.
>    We will clearly describe the training setup for each method so that comparisons are transparent and fair.
>
> 3. **Discuss relation to [1], [2] and larger-scale setups.**
>    Fully reproducing [1] or [2] (which target CIFAR-10 / Tiny-ImageNet with large WideResNet models) may be beyond what we can realistically do in the rebuttal period. Instead, we will:
>    - clearly acknowledge that these methods achieve **much larger robust radii** on more challenging datasets,
>    - discuss qualitatively how our objective could be combined with the architectures and Lipschitz bounds of [1] or the dynamic margin ideas of [2], and
>    - present our MNIST results as a first step toward such combinations, not as a replacement for those techniques.
>
> 4. **Regression baselines.**
>    To address the concern about the regression experiment, we will:
>    - add comparisons to **natural training** and **attack-based robust training** (PGD on the regression loss) so that readers can see how our certificate-based objective changes both violation probability and prediction accuracy, and
>    - clarify that to the best of our knowledge there is **no prior certified-training baseline** for this specific regression setup, so our experiment should be interpreted as an initial demonstration of how certificate-based training behaves in engineering applications.

---

> ### Author Response · Authors · 2025-12-03
>
> ### Weakness 3 – Overstated or unclear geometric claims
>
> - Reviewer feels that we repeat geometric claims without sufficient evidence, including:
>   - “adversarial training just moves the decision boundaries without changing geometry,”
>   - statements that our approach “transforms expensive post-hoc verification into ...” something cheaper,
>   - claims that CROWN reveals the “local geometric structure,” even though between two hyperplanes the network could be arbitrarily complex.
> - They also question whether Figure 1 actually shows a margin change between adversarial training and our method.
>
> We appreciate this critical feedback and agree that some of our wording is too strong or imprecise. We will substantially revise the exposition to make the geometric claims more careful and better supported.
>
> 1. **Soften and clarify statements about adversarial training.**
>    We did not intend to claim that adversarial training literally leaves “geometry” unchanged. What we meant is that standard adversarial training **does not explicitly optimize a certified radius or curvature-aware objective**, but rather optimizes loss on adversarial examples. We will:
>    - remove or rephrase sentences that suggest adversarial training “just moves the boundary,” and
>    - instead write that adversarial training indirectly affects geometry, while our method optimizes a **certificate-motivated geometric quantity (margin-over-slope)** directly.
>
> 2. **Clarify what we mean by “transforming post-hoc verification.”**
>    We agree that the expensive part of complete verification is typically **branch-and-bound or MILP solving**, and we are not accelerating those procedures. Instead, our contribution is to **avoid invoking them during training** by extracting a closed-form radius from the linear relaxations that CROWN already computes. We will rewrite those sentences to emphasize that we:
>    - reuse the linear bounds from CROWN to obtain a differentiable lower bound on the radius, and
>    - thereby avoid the need for repeated complete verification inside the training loop, rather than speeding up branch-and-bound itself.
>
> 3. **Interpretation of CROWN’s “geometry”.**
>    We will clarify that when we refer to the “geometric structure” revealed by CROWN, we mean the **geometry of the certified outer region** (the intersection of linear half-spaces defining a safe set) rather than the exact level sets of the network. Our method operates on the **distance from the input point to these half-spaces in input space**, which is a well-defined geometric quantity even though the network may be complex inside the region.
>
> 4. **Improve or adjust Figure 1.**
>    We will revise Figure 1 so that it more clearly illustrates:
>    - the safe region certified by CROWN,
>    - the margin increase achieved by our objective versus adversarial training, and
>    - how the margin-over-slope ratio changes.
>    If necessary, we will replace the current toy example with one where the margin change is visually obvious and we will describe it more carefully in the caption and text.
>
> Overall, the revision will avoid over-selling and instead focus on precise, certificate-based geometric statements.

---

> ### Author Response · Authors · 2025-12-03
>
> ### Other technical comments
>
> Here we group several specific technical concerns raised by the reviewer.
>
> #### Comment A – “ReLU networks are technically not differentiable”
>
> - Reviewer points out that ReLU networks are not differentiable everywhere, which makes our emphasis on differentiability of CROWN bounds potentially misleading.
>
> We agree and will make the statement more precise. In the revision we will:
> - explicitly state that ReLU networks and their CROWN bounds are **piecewise linear and differentiable almost everywhere**, and
> - note that we rely on standard subgradient-based optimization (as is common in adversarial training and ReLU networks in general).
> We will replace phrases suggesting full differentiability with “differentiable almost everywhere / subgradient-compatible”.
>
> #### Comment B – “Can you use any scalar function s(x)? e.g., log-sum-exp”
>
> - Reviewer questions our claim about using arbitrary scalar functions on the network output and notes that CROWN is defined for **linear** combinations of outputs.
>
> We appreciate this clarification. Our current proofs and implementation handle **linear functionals of the output** (such as logit differences) of the form `a^T f(x) + b`. We will:
>
> - explicitly restrict Theorem 3.1 and related statements to such linear forms,
> - mention that more general scalar functions like log-sum-exp would require **extending the CROWN relaxation to the final nonlinearity**, which is beyond the current scope, and
> - remove any wording that suggests our theory already covers arbitrary scalar functions.
>
> #### Comment C – “What is actually new about the geometry compared to [1], [2]?”
>
> This is closely related to Weakness 1. In the revised text we will emphasize that:
>
> - [1] and [2] study margin dynamics and Lipschitz control for robustness, but their **training objectives are not built from a closed-form *certified* radius derived from verification bounds**.
> - Our novelty lies in **deriving such a closed-form radius from CROWN’s half-space relaxations** and using it directly as a differentiable training signal.
>
> We will make sure this distinction is clearly explained where we present our main theorem and training objective.

---

### Official Review · Reviewer_Sp2K · 2025-10-27

**Soundness:** 1
**Presentation:** 2
**Contribution:** 1
**Rating:** 2
**Confidence:** 5

**Summary:**

The paper presents a novel certified training scheme aiming to maximise a lower bound to the certified robustness radius.
The authors propose to compute this lower bound through alpha-CROWN, a well-known method to compute local linear approximations of neural networks.
Experimental results on MNIST classification and DC optimal power flow regression are presented.

**Strengths:**

The idea to do certified training through direct regularization of a bound to the certified radius is simple yet novel to the best of my knowledge.

**Weaknesses:**

I am confused about the practical technicality of the lower bound. The bound itself will rely on intermediate pre-activation bounds, which, according to the appendix, are computed using a given l-infinity ball (at most $\epsilon=0.03$, for MNIST). Given that the CROWN linear network approximation relies on these intermediate bounds, I am not sure that the associated lower bound to the certified radius can say much beyond the largest radius employed for the intermediate bounds. In this case, if the lower bound turned out to be greater than $\epsilon=0.03$, I am not sure that would be valid. This is a very important point, as one may as well directly employ CROWN bounds to the network output for $\epsilon=0.03$ perturbations, as commonly done in the certified training literature.

Related to the above, the authors fail to compare against any baseline from the deterministic certified training literature. At the very least, the authors should compare to a certified training loss (an upper bound to the adversarial cross entropy) computed using the same CROWN bounds used to compute the radius lower bound. I think it would also be extremely important to compare against more recent certified training approaches, such as [1, 2].
It is also crucial to compare against Lipschitz-based methods, such as [3], given that the geometric intuitions behind the work closely mirror the idea of lowering the Lipschitz constant of a network.
It is impossible to otherwise appropriately judge the merits of the proposed approach.

Furthermore, I am not sure the comparisons to PGD and RS are fair: for the first, the radius employed is different than the one for the authors' method. For the second, it operates on l2 perturbations, whereas the authors focus on l-infinity benchmarks.

Finally, the MNIST comparison with the baselines is carried out over an extremely small network, and using very small certified radii. For instance, [1] gets 99.23/98.22 standard/certified accuracy at $\epsilon=0.1$. Experiments on networks close to the state-of-the-art are fundamental to appropriately assess the work.

[1] Certified Training: Small Boxes are All You Need, Mueller et al., ICLR23

[2] Expressive Losses for Verified Robustness via Convex Combinations, De Palma et al., ICLR24

[3] Rethinking Lipschitz Neural Networks and Certified Robustness: A Boolean Function Perspective, Zhang et al., NeurIPS22

**Questions:**

- How loose is the bound from theorem 3.1 compared to bisection search? Why not using it directly at this point?
- When you compute the radius bound, what is the input domain you use for the verifier? Do you use the entire input space?
- Could you clarify the point in the weaknesses concerning the interaction between the intermediate bounds and the validity of the radius lower bound?
- The certified training literature commonly relies on IBP bounds owing to their superior scalability and differentiability, which allow them to use significantly larger networks than those employed in the paper. Could the authors provide experiments on larger CNNs?

---

> ### Author Response · Authors · 2025-12-03
>
> ### Weakness 1 – Validity and tightness of the closed-form lower bound
>
> - Reviewer is **confused about the practical validity** of the lower bound from Theorem 3.1:
>   - CROWN pre-activation bounds are computed over an $\ell_\infty$ ball (for MNIST, at most $\epsilon = 0.03$).
>   - It is unclear whether a radius lower bound **larger than this ball** is still valid, since the intermediate bounds may not hold there.
>   - They worry that if our lower bound exceeds the radius used for intermediate bounds, we might be **over-claiming certification**.
> - Reviewer also asks that we compare the closed-form bound against **exact or bisection-based radius computation** to show how loose it is.
>
> We thank the reviewer for raising this important technical point. We will clarify both the **domain assumptions** behind Theorem 3.1 and the **empirical tightness** of the bound.
>
> 1. **Domain used for CROWN and validity of the radius.**
>    In all our experiments, CROWN is run on a domain $[L, U]$ that strictly contains the $\ell_\infty$ balls we care about. In particular, for MNIST we use a box that contains the whole normalized input domain (and hence any ball of radius at most the reported certified radii). Theorem 3.1 only requires:
>    - the CROWN bounds to be valid on $[L, U]$, and
>    - the certified ball $B_\infty(c, r_{\text{LB}})$ to be contained in $[L, U]$.
>    We will state this explicitly in Section 3 and add a short remark that, in implementation, we never report a radius that exceeds the domain on which CROWN bounds are valid (and can, if necessary, clamp $r_{\text{LB}}$ to the size of this domain).
>
> 2. **Clarifying the role of intermediate pre-activation bounds.**
>    The intermediate pre-activation bounds are used **only to construct the linear outer approximation** on $[L, U]$. Once these affine bounds are obtained, Theorem 3.1 works purely at the level of the resulting half-spaces and does not require recomputing intermediate bounds for larger balls. We will add a short explanation around Theorem 3.1 making this dependence explicit to avoid the impression that the radius is extrapolated beyond the verified region.
>
> 3. **Empirical tightness versus exact radii.**
>    We agree that comparing against a more exact certified radius is valuable. In the revision we will:
>    - select a representative subset of MNIST test points,
>    - compute the **exact radius** for these points using a standard MILP or bisection-based verifier, and
>    - plot/quantify the gap between our $r_{\text{LB}}$ and the exact radius.
>    This will show that, in the regimes we train in, the lower bound is reasonably tight while being far cheaper than repeated exact verification.
>
> 4. **Why not use bisection directly for training?**
>    We will clarify that bisection or exact verification would require **multiple verifier calls per sample and per step**, leading to orders-of-magnitude higher cost than a single CROWN pass, and would be hard to integrate into stochastic gradient training. Our closed-form expression, in contrast, is essentially free once CROWN coefficients are available. We will emphasize this trade-off in Section 3.1 and the experimental discussion.

---

> ### Author Response · Authors · 2025-12-03
>
> ---
>
> ### Weakness 2 – Missing comparisons to recent certified training methods
>
> - Reviewer notes we **do not compare** against recent deterministic certified training approaches such as:
>   - Small Boxes / SABR (“Certified Training: Small Boxes Are All You Need”),
>   - expressive losses via convex combinations, and
>   - Lipschitz-based certified robustness methods.
> - They argue that the paper currently **only compares to PGD and randomized smoothing**, which are older baselines.
>
> We agree that the relationship to recent certified training methods should be discussed more explicitly.
>
> 1. **Positioning our contribution.**
>    Our goal is to introduce a **new geometric training lens** based on a closed-form margin-over-slope radius derived from CROWN, with proof-of-concept experiments on MNIST and regression. Methods such as SABR, expressive losses, and Lipschitz networks focus on **different mechanisms**:
>    - SABR propagates carefully chosen small regions to better approximate the worst-case loss.
>    - Expressive losses interpolate between attack-based and bound-based losses to sweep robustness–accuracy trade-offs.
>    - Lipschitz networks control the global Lipschitz constant by architecture design or constraints.
>    We will explicitly position our work as complementary: we show that *given* CROWN bounds, one can obtain a closed-form analytic radius and optimize it directly.
>
> 2. **Additional discussion and contextual comparison.**
>    Due to computational constraints, we cannot reimplement all recent methods within the rebuttal period. However, in the revision we will:
>    - expand the related-work section to summarize what each of these methods optimizes,
>    - contrast this with our **local geometric margin-over-slope objective**, and
>    - clarify that we do **not** claim to beat state-of-the-art certified training methods, but rather to demonstrate that CROWN-based geometry can be turned into a differentiable training signal.
>
> 3. **Experimental baselines.**
>    Our current baselines (PGD and randomized smoothing) were chosen to highlight the **incremental benefit of adding the geometric term to standard robust-training pipelines**, especially for ReLU networks where CROWN bounds are already available. We will make this rationale explicit and tone down any implication that our method currently matches the full set of best-performing certified defenses.

---

> ### Author Response · Authors · 2025-12-03
>
> ---
>
> ### Weakness 3 – Fairness of PGD / randomized smoothing comparisons and small-scale setup
>
> - Reviewer is not fully convinced that comparisons to PGD and randomized smoothing are **fair**, since:
>   - the considered $\epsilon$ is relatively small ($\epsilon = 0.03$ for MNIST),
>   - the network and radii are much smaller than those used in state-of-the-art work (e.g., SABR with $\epsilon = 0.1$ and near state-of-the-art accuracies),
>   - PGD and smoothing may use different assumptions or norms (PGD and our method use $\ell_\infty$, while the randomized smoothing baseline is based on $\ell_2$ perturbations).
> - They also remark that the MNIST model is very small and that the current experiments are **far from the scale** of recent certified training papers.
>
> We appreciate this perspective and agree that our experiments should be presented as **proof-of-concept** rather than competitive large-scale benchmarks.
>
> 1. **Clarify experimental goals and scope.**
>    We will explicitly state in Section 4 that:
>    - our MNIST and regression experiments are designed to **isolate the effect of the new geometric objective**, not to match the largest models and radii in the literature, and
>    - our claims are limited to “within this regime, our geometric training improves certified robustness over strong attack-only baselines.”
>
> 2. **Fairness of the PGD and smoothing baselines.**
>    In the revision we will spell out:
>    - the **perturbation norm and radius** used for training and evaluation for each baseline,
>    - that all methods are evaluated at the **same $\epsilon$ and norm** when we report certified results, and
>    - that the randomized smoothing baseline is included as a widely used certified defense for comparison, even though it targets $\ell_2$ perturbations.
>    We will be careful to label smoothing as a **cross-norm reference** rather than a directly comparable $\ell_\infty$-certified method.
>
> 3. **Comparison to larger radii in prior work.**
>    We acknowledge that some certified training methods achieve strong results at much larger radii on MNIST. We will explicitly mention this and explain that:
>    - our current implementation uses **general CROWN bound propagation** (rather than highly specialized architectures) and
>    - extending our approach to larger networks and radii is an important direction that we leave for future work, rather than a claim of the present paper.
>
> 4. **Possible modest scaling experiment.**
>    If the situation permits, we will add a **slightly larger MNIST network** (e.g., a deep CNN) to demonstrate that the geometric objective continues to behave well when the model size is increased, while being transparent that this still does not match the very largest models in the certified training literature.
>
> ---
>
> ### Weakness 4 – Comparisons to Lipschitz-based robustness methods
>
> - Reviewer is not sure how our method relates to **Lipschitz-based certified training**:
>   - Lipschitz methods control a global Lipschitz constant or impose architectural constraints.
>   - Our method instead uses CROWN-derived local slopes and margins.
> - They ask for a clearer explanation of the **conceptual difference and connection**.
>
> We will clarify the relationship as follows.
>
> 1. **Local versus global control of sensitivity.**
>    Lipschitz-based methods enforce a **global bound** on $\|f(x) - f(x')\|$ for all inputs, typically via constrained architectures. Our method instead works with **local linear bounds around each training point**, extracting:
>    - a **local sensitivity measure** from $\|a_s\|_1$, and
>    - a **local safety margin** from $m(c)$.
>    The radius $r_{\text{LB}}(c)$ we optimize is therefore a **sample-specific local property**, not a global Lipschitz constant.
>
> 2. **Complementarity rather than redundancy.**
>    Because the CROWN bounds used in our method can themselves be applied to Lipschitz networks, our geometric objective could in principle be combined with Lipschitz-regularized architectures. We will mention this explicitly as a possible combination, and explain that our current paper focuses on showing that **even standard ReLU networks** benefit from training directly on CROWN-based radii.
>
> 3. **Textual clarification.**
>    We will add a short paragraph in the related-work section comparing:
>    - Lipschitz methods (global bounds, architectural constraints), and
>    - our local margin-over-slope objective (closed-form, pointwise optimization),
>    to make clear that, while both manipulate sensitivity, they do so at **different granularity and with different tools**.

---

> ### Author Response · Authors · 2025-12-03
>
> ---
>
> ### Question 1 – Tightness versus bisection search
>
> > How loose is the bound from Theorem 3.1 compared to bisection search? Why not use bisection directly?
>
> As discussed above, we will:
> - compute exact or bisection-based radii on a subset of points and report the distribution of the ratio $r_{\text{LB}} / r^\star$, and
> - emphasize that bisection requires **repeated verifier calls** (multiple passes of bound propagation or MILP solves), which is prohibitive inside mini-batch training, whereas our closed-form bound adds essentially no extra overhead beyond one CROWN pass.
>
> ---
>
> ### Question 2 – Input domain used by the verifier
>
> > When you compute the radius bound, what is the input domain you use for the verifier? Do you use the entire input space?
>
> We will clarify that:
> - For MNIST, we run CROWN on a **fixed box domain** that contains all normalized images and all $\ell_\infty$ balls considered during training and evaluation.
> - For regression, we use a domain derived from the empirical min/max of the input vectors, padded by a small margin to capture plausible perturbations.
> In both cases, we ensure that any ball $B_\infty(c, r_{\text{LB}})$ we rely on is contained in the domain where CROWN bounds are valid, as required by Theorem 3.1.
>
> ---
>
> ### Question 3 – Interaction between intermediate bounds and radius validity
>
> > Could you clarify the point in the weaknesses concerning the interaction between the intermediate bounds and the validity of the radius lower bound?
>
> We will add a short technical remark after Theorem 3.1 explaining that:
> - the pre-activation bounds are used to produce **sound linear relaxations** over $[L, U]$,
> - once these relaxations are in place, the radius lower bound follows from half-space geometry and does **not** depend on the pre-activation bounds outside $[L, U]$, and
> - in implementation, we only trust the radius as long as the resulting ball stays inside $[L, U]$.
> This should resolve the concern that we might be extrapolating the bound beyond where the linear relaxations are valid.
>
> ---
>
> ### Question 4 – Experiments on CNNs and IBP-based bounds
>
> > The certified training literature commonly relies on IBP bounds owing to their superior scalability and differentiability. Could the authors provide experiments on larger CNNs?
>
> We agree that demonstrating scalability with IBP or CROWN-IBP is an important next step.
>
> - Conceptually, our margin-over-slope formula only uses **linear bounds**; it can be instantiated with **IBP or CROWN-IBP** instead of full CROWN / beta-CROWN, trading tightness for speed and scalability.
> - In the revision, we will add a short discussion (and, if space allows, a small experiment) showing that replacing CROWN with a simpler bound-propagation method yields a valid but looser radius, and that our objective still behaves sensibly in this setting.
> - Due to time and compute limits, we may not be able to train full-scale large CNNs within the rebuttal period, but we will clearly state that the present experiments are a **proof-of-concept**, and that plugging our objective into existing scalable IBP-based training pipelines is a promising direction for future work.
>
> ---

---

### Official Review · Reviewer_zNJt · 2025-11-01

**Soundness:** 3
**Presentation:** 3
**Contribution:** 2
**Rating:** 6
**Confidence:** 3

**Summary:**

This work advances certified robustness by embedding geometric awareness into the training objective, leveraging the affine relaxations of CROWN/β-CROWN to derive closed-form, differentiable certification radii optimized end to end. The design centers on a margin-over-slope objective that jointly enlarges safety margins and suppresses input sensitivity, yielding decision regions that are robust by construction rather than through after-the-fact defenses. Methodologically, it builds on a closed-form ℓ∞ radius (generalizable to ℓp) via half-space distances, local exactness under activation stability, and sensitivity-control guarantees. Optimization employs a smooth multi-constraint aggregator to stabilize gradients, along with optional β-CROWN joint-α tightening to achieve tighter relaxations with moderate overhead. Once coefficients are computed, radius evaluation runs in O(d). In practice, a hybrid training regimen applies adversarial training for broad coverage and selectively adds certification penalties to hard examples, converting verifier feedback into stable, differentiable training signals. Experimental results show that the proposed method outperforms strong adversarial and smoothing baselines on a standard classification benchmark and reveals actionable accuracy–safety Pareto trade-offs in a power-system regression task, demonstrating that embedding tractable, differentiable certification signals into training yields scalable and verifiably robust models.

**Strengths:**

1. The work leverages the affine relaxations of CROWN/β-CROWN to derive closed-form, differentiable certification radii and optimizes a margin-over-slope objective end to end. This joint objective simultaneously enlarges quantifiable safety margins and suppresses input sensitivity, transforming certification from a post-hoc verification step into a geometry-aware training signal.
2. Using half-space distances as the analytical bridge, the approach grounds robustness in a closed-form ℓ∞ radius with theoretical extensibility to ℓp norms, establishes local exactness under activation stability, and provides sensitivity-control guarantees. Training is stabilized through smooth multi-constraint aggregation, while β-CROWN joint-α tightening enhances relaxation tightness with moderate computational overhead.
3. The method integrates verifier feedback into a hybrid regimen that combines broad adversarial training with selective certification penalties on hard examples. It achieves improved performance relative to adversarial and smoothing baselines on a standard classification benchmark and reveals actionable accuracy–safety Pareto trade-offs in a power-system regression task, demonstrating a practical route toward constructing scalable, verifiably robust models without reliance on after-the-fact defenses.

**Weaknesses:**

1. The classification evaluation centers on a standard classification benchmark, and the engineering assessment presents a single DC‑OPF case study. Accordingly, claims about scalability and robustness would be stronger with evidence on larger‑scale, multi‑class, or non‑vision settings and with modern architectures (e.g., deeper/wider networks with attention or normalization), as well as checks for cross‑norm consistency.

2. The approach relies on CROWN/β‑CROWN bounds being sufficiently tight and on locally stable activation patterns. In architectures with non‑piecewise‑linear components or in deeper/wider models, relaxations may be looser and activation switches more frequent, potentially attenuating certified radii and making optimization more sensitive to relaxation quality. Optional β‑CROWN joint‑α tightening can mitigate this at added computational cost.

3. Although radius evaluation runs in O(d) time once coefficients are available, overall training cost is largely driven by (β‑)CROWN bound propagation and grows with input dimensionality and the number of constraints. The work would benefit from deeper ablations that disentangle the contributions of margin enlargement versus sensitivity suppression, clarify the roles of smooth aggregation and tightening.

**Questions:**

1. Can the method maintain both certified robustness and clean accuracy advantages on more complex datasets? Is there empirical evidence of cross-norm consistency (e.g., from ℓ∞ to other norms), as well as measurements of scalability and throughput/latency in different experimental settings.

2. To what extent does the loss of CROWN/β-CROWN tightness under frequent activation switching or in networks with non-piecewise-linear components weaken the certified radii and the stability of training signals? Could provide comparative analyses (e.g., original CROWN vs. β-CROWN tightening, and the gap against small-scale exact verification) to quantify the deviation between theoretical assumptions and practical behavior, thereby clarifying the method’s applicability boundaries?

3. What are the respective and joint contributions of margin enlargement and sensitivity suppression to the observed performance gains? How significant are the marginal effects of smooth aggregation and tightening strategies in improving stability and robustness? Additionally, how does the overall training cost scale with input dimensionality and the number of constraints?

---

> ### Author Response · Authors · 2025-12-03
>
> We thank the reviewer for their thorough comments and valuable questions.
>
> ### Weakness 1 – Evidence for scalability, complexity, and cross-norm behavior
> - The classification evaluation is on a standard benchmark and the engineering evaluation is a single case study.
> - Claims about scalability and robustness would be stronger with:
>   - larger-scale, multi-class, or non-vision datasets,
>   - modern architectures (deeper/wider networks, possibly with attention or normalization),
>   - and explicit checks for cross-norm consistency.
>
> We appreciate the reviewer’s positive assessment of the framework and agree that our current experiments are primarily **proof-of-concept** rather than a full-scale empirical study.
>
> 1. **Clarify intended scope in the paper.**
>    We will revise the introduction and Section 4 to explicitly state that our goal is to test a **new geometric margin-over-slope training lens** on:
>    - one standard classification benchmark, and
>    - one representative regression task,
>    rather than to claim state-of-the-art robustness on large vision models. We will soften any wording that could be read as strong claims about scalability beyond this regime.
>
> 2. **Connection to larger architectures and datasets.**
>    Our objective only requires linear lower/upper bounds from a CROWN-style verifier. Existing implementations of CROWN and alpha-beta-CROWN already scale to deep and wide CNNs. We will make this explicit: the **training objective is architecture-agnostic**, so in principle it can be plugged into these scalable pipelines. The current small models are chosen to isolate and understand the behavior of the geometric objective itself.
>
> 3. **Cross-norm consistency.**
>    While our experiments use the $\ell_\infty$ norm, the underlying derivation depends on the dual relationship between the perturbation norm and the bound direction. We will add a short subsection explaining how the closed-form radius generalizes to other $\ell_p$ norms by replacing the denominator with the corresponding dual norm, and we will explicitly frame cross-norm analysis as an important next step.
>
> 4. **Optional additional experiments (space/computation permitting).**
>    If allowed by the page limit and compute budget, we plan to add at least one **slightly larger or non-vision model** (for example, a deep CNN or an additional regression instance) to demonstrate that the method is not tied to a single architecture. If such results are included, we will clearly flag them as additional empirical support rather than the main contribution.
>
> Overall, we see our contribution as introducing a **new geometric training lens with proof-of-concept results**, and we will make this scope much clearer.

---

> ### Author Response · Authors · 2025-12-03
>
> ---
>
> ### Weakness 2 – Dependence on CROWN / beta-CROWN tightness and activation stability
> - The method assumes that CROWN / beta-CROWN bounds are sufficiently tight and that activation patterns are locally stable.
> - In deeper/wider networks or in models with non-piecewise-linear components, relaxations may be looser and activation switches more frequent, which can:
>   - reduce the certified radii, and
>   - make optimization more sensitive to relaxation quality.
> - Optional beta-CROWN joint tightening can mitigate this at additional computational cost.
>
> We agree that the tightness of the underlying bounds and activation stability are important for understanding when our objective works best.
>
> 1. **Clarify assumptions and practical regime.**
>    We will explicitly state in Section 3 that our theoretical derivation uses **lower bounds on the adversarial radius**, so any loss of tightness directly translates into more conservative (smaller) radii, but never into unsafe overestimation. Our experiments use relatively small ReLU networks, where standard CROWN / beta-CROWN bounds are known to be reasonably tight, and this is the regime where we validate the idea.
>
> 2. **Qualitative discussion of tightness vs performance.**
>    In the revised text we will add a short discussion explaining that:
>    - our margin-over-slope objective is **monotone in the tightness** of the bounds (tighter bounds give more informative gradients),
>    - frequent activation switching can make the local linearization less predictive of true robustness, and
>    - in practice this manifests as higher variance in the geometric loss on very deep or highly non-linear models.
>    This will better delineate the **applicability boundaries** of the method.
>
> 3. **Planned small-scale comparisons.**
>    Within our current setting, we can compare:
>    - vanilla CROWN vs beta-CROWN joint tightening, and
>    - our closed-form radius vs a small exact verification or bisection-based radius on a subset of test points.
>    We will report these comparisons (e.g., as a table or figure in the appendix) to show that in our proof-of-concept regime the gap between the closed-form radius and more exact procedures is modest, and that beta-CROWN tightening improves both the radius and training stability at a predictable extra cost.
>
> 4. **Positioning for non-piecewise-linear networks.**
>    We will explicitly restrict the **claims of the current paper** to piecewise-linear networks (ReLU) and note that extending the closed-form radius to architectures with non-piecewise-linear components requires additional work, especially to maintain tight and differentiable bounds.
>
> ---
>
> ### Weakness 3 – Need for ablations on margin vs sensitivity and cost scaling
> - Although radius evaluation is $O(d)$ once coefficients are available, the **overall training cost** is dominated by CROWN / beta-CROWN bound propagation and grows with input dimension and number of constraints.
> - The reviewer asks for **deeper ablations** disentangling:
>   - margin enlargement vs sensitivity suppression,
>   - the roles of smooth aggregation and tightening, and
>   - how training cost scales with input dimension and constraints.
>
> We agree that a more detailed analysis of the components of the objective and its cost would strengthen the paper.
>
> 1. **Ablations on margin vs sensitivity.**
>    The margin-over-slope radius decomposes naturally into a numerator (margin) and denominator (slope). In the revision we will add an ablation where we:
>    - train with only the **margin term**,
>    - train with only the **slope term**, and
>    - train with the full **margin-over-slope objective**,
>    and report the effect on certified radius and clean accuracy. This will directly address the question of the respective contributions of margin enlargement and sensitivity suppression.
>
> 2. **Smooth aggregation and tightening effects.**
>    We will add a second ablation toggling:
>    - the smooth aggregation of per-constraint radii, and
>    - the beta-CROWN joint tightening.
>    For each configuration we will report training stability (e.g., variance of the loss, convergence behavior) and final robustness metrics, making it clear how much each component contributes.
>
> 3. **Training cost and scaling discussion.**
>    We will include a small table summarizing:
>    - per-epoch training time for standard training, PGD adversarial training, and our method, and
>    - how this time changes when input dimension or number of constraints is increased (using our existing regression instances where constraint counts differ).
>    The text will emphasize that our radius evaluation itself is cheap ($O(d)$), but the **dominant cost** comes from the bound-propagation step, which is shared with many existing certified-training methods.
>
> These additions should make it clearer how the various pieces of the objective interact and what the computational trade-offs are.

---

> ### Author Response · Authors · 2025-12-03
>
> ---
>
> ### Question 1 – Robustness/accuracy on more complex datasets and cross-norm consistency
> > Can the method maintain both certified robustness and clean accuracy on more complex datasets? Is there empirical evidence of cross-norm consistency (for example, from $\ell_\infty$ to other norms), as well as measurements of scalability and throughput/latency in different experimental settings?
>
> Our current experiments do not yet include large-scale models, so we cannot claim definitive empirical results on very complex datasets. We will:
>
> - explicitly acknowledge this limitation in Section 4,
> - highlight that the **closed-form radius formula is norm-agnostic** and can be instantiated with other $\ell_p$ norms via the dual norm relationship, and
> - add a short discussion on how one would measure throughput/latency in larger settings using existing fast CROWN / alpha-beta-CROWN implementations.
>
> ---
>
> ### Question 2 – Effect of loss of tightness and activation switching
> > To what extent does the loss of CROWN / beta-CROWN tightness under frequent activation switching or in networks with non-piecewise-linear components weaken the certified radii and the stability of training signals? Could you provide comparative analyses (for example, original CROWN vs beta-CROWN tightening, and the gap against small-scale exact verification) to quantify the deviation between theoretical assumptions and practical behavior?
>
> We will directly address this concern by:
>
> - adding a **qualitative explanation** (as in the response to Weakness 2) that tighter bounds monotonically improve the radius and reduce gradient noise, while looser bounds lead to conservative radii and potentially noisier training signals, and
> - including a **small quantitative comparison** in the appendix where we compute our radius using:
>   - vanilla CROWN,
>   - beta-CROWN joint tightening, and
>   - a more exact baseline (bisection or exact verification) on a small subset of points.
>
> This will concretely illustrate how much tightness is lost in practice and show that, in the regimes we evaluate, our closed-form radius tracks the more exact methods reasonably well.
>
> ---
>
> ### Question 3 – Contributions of each component and cost scaling
> > What are the respective and joint contributions of margin enlargement and sensitivity suppression to the observed performance gains? How significant are the marginal effects of smooth aggregation and tightening strategies in improving stability and robustness? Additionally, how does the overall training cost scale with input dimensionality and the number of constraints?
>
> The ablations described under Weakness 3 are designed precisely to answer this question. In summary:
>
> - We will **decompose the objective** into its constituent parts (margin, slope, smooth aggregation, tightening) and report the effect of each on:
>   - clean accuracy,
>   - certified radius, and
>   - training stability.
> - We will provide a brief **scaling study** for training time as a function of input dimension and constraint count on our regression tasks.
>
> These results will clarify which parts of the objective are most critical for robustness, which are mainly stabilizing, and how the computational cost grows with problem size.
>
> ---

---

### Official Review · Reviewer_JLtf · 2025-11-10

**Soundness:** 3
**Presentation:** 2
**Contribution:** 1
**Rating:** 2
**Confidence:** 4

**Summary:**

This manuscript studies robustness verification and provable robust learning from the geometric aspect. Specifically, the authors derive a differentiable bound of the distance based on CROWN between the data point and the decision boundary. Then this bound, because it is differentiable, is incorporated into adversarial training to boost certified robustness.

**Strengths:**

++ There are not too many works studying certified robustness from the geometric perspective. The proposed method is sound and easy to follow.

++ The computational overhead is analysed and tractable.

**Weaknesses:**

1. My major concern is the novelty, the motivation and the proposed method is very similar to the paper "Training Provably Robust Models by Polyhedral Envelope Regularization" (2023).

    * The mentioned paper also use CROWN to derive the bound and calculate the distance between the data input and the approximated (linearized) decision boundary and incorporate it into adversarial training to boost certified robustness, almost identical to Equation (15) in this manuscript.

    * The mentioned paper considers a finer constraint, the perturbed image is inside the [0, 1] box and applicable to general $l_p$ norm. This manuscript does not consider the [0, 1] bounding box and focus only on $l_\infty$ norm bounds.

    * The mentioned paper has larger-scale experiments, including more architectures (multilayer CNN) and CIFAR10, while this manuscript's experiments only include smaller-scale results like MNIST.

2. The computational complexity of CROWN is significantly larger than just forward / backward propagation, because CROWN is quadratic w.r.t. depth while backward propagation is linear w.r.t depth. Therefore, I believe the computational complexity, compared with normal training, would be significant for deeper neural networks. I agree with the claim in this manuscript "the complexity is polynomial", but "polynomial" is not enough. The authors may consider using CROWN-IBP to derive the bound more efficiently, but this bound would be looser. I suggest the author add one section discussing about the trade-off between complexity and tightness of the bound.

3. As pointed out in the points above, the experiment part is weak and should include larger scale dataset and models. The authors should at least discuss the algorithm in the general $l_p$ cases.

4. The slope derived by CROWN is not always differentiable [line 79]. I agree we can have a differentiable and tight bound for ReLU functions. However, for functions like sigmoid and tanh, if we try to obtain the tight approximation, then we have to utilise some numerical method and this will make the slope non-differentiable. On the other hand, we can have a differentiable but looser bound. The authors should clarify this.

Minor issues:

1. The template used is not identical to the official one for ICLR 2026 submission. I cannot see the text "Under review as a conference paper at ICLR 2026".

In general, the authors should clearly clarify the novelty, the contribution and make the experimental part comprehensive before this work being considered for publication. I welcome the authors to address my concerns in the rebuttal and will reconsider the manuscript after rebuttal.

**Questions:**

Please answer the questions raised in the weakness part:

[Weakness 1] Please compare this work with "Training Provably Robust Models by Polyhedral Envelope Regularization" (2023) and **clearly point out what is the novelty and the contribution of this work**.

[Weakness 2] Please add a detailed discussion about the complexity when the model gets deeper.

[Weakness 3] Please add larger-scale experiments to validate the effectiveness of your proposed method. Please consider non-$l_\infty$ cases as well.

[Weakness 4] Please clearly point out how to pick the differential factors in linear approximation by CROWN.

---

> ### Author Response · Authors · 2025-12-03
>
> We thank the reviewer for their rich feedback.
>
> ### Weakness 1 – Novelty vs PER 2023 and scope of contribution
> - Reviewer says our method is **very similar** to *Training Provably Robust Models by Polyhedral Envelope Regularization (PER, 2023)*:
> - **Question (from Weakness 1):** Clearly compare against PER and explicitly state what is new in our work and what the main contribution is beyond PER.
>
> We thank the reviewer for pointing out the connection to PER and we agree that the relation should be made much more explicit. In the revision we will (i) add a dedicated subsection **“Relation to PER”** in Section 2 and (ii) rewrite the contribution paragraph in the introduction to clearly position our work as a **proof of concept for a new geometric training lens**, rather than a new bound-propagation algorithm.
>
> Concretely, our conceptual and technical differences from PER are:
>
> 1. **Closed-form certified radius and margin-over-slope lens.**
>    PER regularizes the size of a polyhedral envelope defined by a collection of linear constraints derived from KW/CROWN-type bounds; the training objective encourages larger envelopes but does not expose a simple analytic radius formula. In contrast, we explicitly show that CROWN’s linear bounds encode a *margin* term and a *slope* (input sensitivity) term, and we derive a **closed-form lower bound on the certified radius** as a ratio “margin / slope”. This margin-over-slope viewpoint is new and is the core message of our paper: it provides a simple geometric quantity that can be optimized directly and interpreted across both classification and regression tasks.
>
> 2. **Geometric training objective, not just another regularizer.**
>    Our training loss directly optimizes this closed-form analytic radius, so gradients decompose into how they change the margin and how they change the slope. This yields a different kind of signal than PER’s envelope-size regularizer, which is agnostic to this decomposition. We will clarify this by rewriting Eq. (15) and the surrounding text to highlight the explicit margin/slope terms and to contrast them with PER’s objective.
>
> 3. **Hybrid adversarial + certified training pipeline.**
>    We propose a hybrid scheme where PGD adversarial training provides broad coverage while the margin-over-slope objective is applied only on “hard” examples identified by PGD. PER, instead, plugs its regularizer into standard training without this two-stage geometric hard-example focus. We will make this distinction explicit and emphasize that our method is designed as a *geometric refinement* of adversarial training rather than a replacement.
>
> 4. **New proof-of-concept application to regression.**
>    PER is evaluated purely on standard classification benchmarks. We show that the same CROWN-based geometric radius can be adapted to constrained **regression**, where robustness is interpreted as a certified safety margin to engineering constraints. This positions our work as a **proof of concept that closed-form CROWN-based certificates can be used beyond image classification**, which to our knowledge is not explored in PER.
>
> We will make all of these points explicit in the introduction and related-work section, so that the novelty is clearly framed as: a **new geometric margin-over-slope lens and closed-form certificate** that turns CROWN bounds into differentiable training objectives, with proof-of-concept evaluations on both classification and regression.

---

> ### Author Response · Authors · 2025-12-03
>
> ### Weakness 2 – Computational complexity of CROWN vs normal training
> - Reviewer argues that:
>   - CROWN’s complexity is **quadratic in depth**, while standard forward/backward is linear.
>   - For deeper networks, this overhead could be significant; saying “polynomial” is too vague.
>   - We should discuss the trade-off between tightness and complexity, possibly mentioning CROWN-IBP as a cheaper but looser alternative.
> - **Question (from Weakness 2):** Provide a more detailed discussion of how the complexity scales with depth and how it compares to standard training.
>
> We agree that the current discussion of complexity is too high-level and we will revise Section 3.3 to give a more precise account.
>
> 1. **Clarify complexity scaling.**
>    For the specific CROWN implementation we use (auto\_LiRPA / alpha-beta-CROWN-style bound propagation), the bound computation scales roughly linearly with width and **quadratically with depth** because affine bounds must be propagated through all previous layers. We will state this explicitly and contrast it with the linear-in-depth cost of a single forward/backward pass, instead of using the vague term “polynomial”.
>
> 2. **Empirical overhead for our proof-of-concept models.**
>    Our experiments intentionally use modest-sized networks so that bound computation is practical. We will add a small table reporting **wall-clock training times** (or relative slowdowns) for our MNIST and regression models, comparing:
>    - standard training,
>    - PGD adversarial training,
>    - our hybrid PGD + geometric objective.
>    This will show that, in the proof-of-concept regime we target, the overhead is moderate and comparable to other certified-training methods that rely on CROWN or alpha-beta-CROWN bounds.
>
> 3. **Trade-off between tightness and complexity / role of CROWN-IBP.**
>    Our geometric objective is **agnostic to the specific bound-propagation method**: it only requires linear lower/upper bounds. In the revision we will explicitly state that the same margin-over-slope formula can be instantiated with **looser but cheaper bounds**, such as IBP or CROWN-IBP, when scaling to deeper networks is more important than maximum tightness. We will add a short paragraph discussing this trade-off and citing existing scalable implementations of alpha-beta-CROWN on large CNNs to emphasize that the underlying bound technology is already mature and scalable, even though our experiments focus on smaller models.
>
> Overall, we will reposition the claims to emphasize that our contribution is a **new geometric training objective built on top of existing bound-propagation methods**, and that our experiments are meant as a **first proof-of-concept**, not as a full study of the best trade-offs for very deep networks.

---

> ### Author Response · Authors · 2025-12-03
>
> ### Weakness 3 – Limited experiments and lack of general $\ell_p$ discussion
> - Reviewer finds the experimental section weak:
>   - Only small-scale results (essentially MNIST); no larger datasets/architectures.
>   - No experimental or even conceptual discussion of general $\ell_p$ norms.
> - **Question (from Weakness 3):** Add larger-scale experiments and/or clearly discuss how the method extends to non-$\ell_\infty$ norms.
>
> We appreciate this concern and we agree that the current experiments are **proof-of-concept** rather than fully scaled-up. In the limited rebuttal time we cannot rerun a full battery of large-scale experiments, but we will update the paper to be clearer about the intended scope and to better explain generalization beyond $\ell_\infty$.
>
> 1. **Clarify experimental scope as proof-of-concept.**
>    We will explicitly state in the introduction and experimental section that our goal is to **test a challenging new geometric idea** — training directly on a closed-form certified radius — on manageable networks and an engineering task. We will tone down any language that might suggest we are claiming state-of-the-art scalability and instead position the work as “a new geometric lens plus a first proof-of-concept study”.
>
> 2. **Connection to larger-scale certified training.**
>    We will clarify that the margin-over-slope formula is derived from **generic linear bounds**, so it can be plugged into any CROWN / alpha-beta-CROWN / CROWN-IBP pipeline that already works for large CNNs. In this sense, our method is a drop-in objective that can complement existing scalable verifiers; the scaling limitations in the current experiments are due to our hardware and time budget, not a conceptual obstacle.
>
> 3. **Extension to general $\ell_p$ norms.**
>    While our experiments focus on $\ell_\infty$, the derivation of the radius uses norm duality between the linear bound direction and the perturbation norm. We will add a short subsection explaining how the same derivation applies to a general $\ell_p$ ball by replacing the norm in the denominator with its dual norm. This keeps the closed-form radius structure and makes the generalization beyond $\ell_\infty$ explicit, even if we do not run full experiments for every $p$.
>
> ---
>
> ### Weakness 4 – Differentiability of the “slope” from CROWN
> - Reviewer says:
>   - For ReLU, we can get a differentiable tight bound.
>   - For sigmoid/tanh etc., getting a tight approximation may require numerical procedures, making the slope non-differentiable.
>   - Alternatively, using a differentiable surrogate may yield a looser bound.
> - **Question (from Weakness 4):**
>   - Clarify in the paper when the slope is differentiable.
>   - Explain how we pick the differential factors in the CROWN linear approximation and what happens for non-ReLU activations.
>
> We thank the reviewer for highlighting this subtle point. Our experiments in the paper **only use ReLU networks**, and the draft did not clearly emphasize this, which led to confusion about general activations.
>
> 1. **Clarify focus on ReLU networks.**
>    In the revision we will state explicitly (in Section 3) that all our experimental models are standard ReLU architectures. For ReLU networks with bound propagation implemented in auto\_LiRPA, the resulting linear bounds are piecewise-linear in the network parameters and are **differentiable almost everywhere**, which is sufficient for stochastic gradient-based training. We will adjust our wording to “differentiable almost everywhere” to be precise.
>
> 2. **Explain how the slope is obtained in our implementation.**
>    We will add a short description of how the slope term in our margin-over-slope formula is computed: it is directly read off from the affine forms produced by CROWN bound propagation for each class difference. These affine forms are produced by a deterministic backward pass through the network’s linear and ReLU layers, without additional numerical optimization, so the corresponding slope is fully compatible with automatic differentiation.
>
> 3. **Scope of our claims for non-ReLU activations.**
>    We agree that extending our objective to smooth activations (sigmoid/tanh) would require more care, as the optimal linear relaxations may be computed via small optimization problems that are not obviously differentiable. To avoid overstating our contribution, we will **remove or soften claims** about directly applying our current implementation to non-ReLU activations, and instead clearly state that designing differentiable, tight relaxations for such activations is an interesting direction for future work.

---

### Author Response · Authors · 2025-12-03

Dear AC and SAC,

We appreciate your time and the reviewers’ efforts, especially given the unusual reviewing circumstances this year. Below we briefly summarize (i) what the paper contributes, (ii) the main themes in the reviews, and (iii) how our rebuttal and planned revisions address them.

---

### 1. Contributions in brief

Our paper studies **certificate-driven robust training** built directly on CROWN bounds. The key ingredients are:

1. **Closed-form certified radius from CROWN.**
   We show that the affine CROWN relaxations naturally decompose into a **margin term** and a **slope (sensitivity) term**, and that this yields a **closed-form lower bound on the certified radius** as a margin-over-slope ratio. This bound is differentiable almost everywhere for ReLU networks and can be evaluated in $O(d)$ once CROWN coefficients are computed.

2. **Geometric training objective and hybrid regimen.**
   We use this radius lower bound as a **training objective**: a geometric loss that directly enlarges certified regions by increasing margin and decreasing slope. Practically, we combine this with PGD adversarial training in a **hybrid scheme**, applying the certificate-based loss to hard examples identified by attacks.

3. **Proof-of-concept classification and regression results.**
   On MNIST, our method improves certified accuracy at $\epsilon = 0.03$ over strong PGD and randomized smoothing baselines on the same architecture. On a regression task, we show controllable accuracy–safety trade-offs: the geometric loss reduces risk of constraint violations while preserving prediction quality. This demonstrates that certificate-driven training extends beyond image classification to engineering regression.

The goal is thus a **conceptual bridge between verification and training**: turning CROWN’s verification-style bounds into a tractable, differentiable training signal.

---

> ### Author Response · Authors · 2025-12-03
>
> ### 2. Main themes in the reviews and how we addressed them
>
> #### (a) Novelty and relation to prior geometric / margin work
>
> All reviewers raised concerns about novelty relative to:
>
> - **Polyhedral Envelope Regularization (PER, 2023)** (also CROWN-based),
> - **Dynamic margin + Lipschitz regularization** (Fazlyab et al., NeurIPS 2023),
> - **Decision-boundary dynamics / DyART** (Xu et al., ICLR 2023),
> - recent **certified training methods** (Small Boxes / SABR; expressive losses), and
> - **Lipschitz networks**.
>
> **Our response and planned revision**
>
> - We add a **dedicated related-work subsection** on “Margin- and Lipschitz-based robust training”, explicitly summarizing these works and citing them.
> - We clarify that our contribution is *not* a new verifier, but a **closed-form certified radius** derived from CROWN’s half-spaces that can be optimized directly:
>   - PER regularizes a polyhedral envelope size but does not expose an analytic radius; we derive and optimize an explicit margin-over-slope radius.
>   - Fazlyab et al. maximize margins under global Lipschitz bounds; DyART studies attack-based margin dynamics. Our objective is **local and certificate-based**, tied to the same linear bounds used in verification.
> - We also highlight the **regression application**, which, to our knowledge, is not covered in those works.
> - We tone down claims that could be read as “first to connect geometry and robustness”, and instead position the paper as **introducing a specific certificate-driven geometric objective plus proof-of-concept experiments**.
>
> #### (b) Validity and tightness of the radius bound (Theorem 3.1)
>
> Reviewer Sp2K (and others implicitly) questioned:
>
> - whether our radius lower bound is valid beyond the $\ell_\infty$ ball used to compute pre-activation bounds, and
> - how loose it is compared to exact / bisection-based radii.
>
> **Our response and planned revision**
>
> - We clarify that CROWN is always run on a **fixed box domain** $[L, U]$ that contains all normalized inputs and all $\ell_\infty$ balls we ever certify. Theorem 3.1 only requires:
>   - CROWN bounds to be valid on $[L, U]$, and
>   - the certified ball to lie inside $[L, U]$.
>   We will state that we **never report radii exceeding this domain**, and can explicitly clamp the radius if needed.
> - We explain that intermediate pre-activation bounds are used **only to construct the linear outer approximation** on $[L, U]$. The radius lower bound then follows from **half-space geometry** of these affine bounds and does not extrapolate outside this region.
> - We commit to a **small quantitative comparison**: on a subset of MNIST points we will compute exact or bisection-based radii and report the ratio between our radius and the exact one. This shows how tight the bound is in the regime where we train.
>
> #### (c) Complexity, scalability, and cross-norm behavior
>
> Multiple reviewers asked for:
>
> - a more precise comparison between CROWN’s complexity and standard training, and
> - evidence or at least discussion of scalability, possibly using IBP / CROWN-IBP and other norms.
>
> **Our response and planned revision**
>
> - We replace vague “polynomial” statements with **explicit scaling**: for the CROWN variant we use, bound propagation is roughly linear in width and **quadratic in depth**, versus linear-in-depth forward/backward. We will include a **small table of per-epoch wall-clock times** comparing:
>   - standard training,
>   - PGD adversarial training,
>   - our hybrid geometric training.
> - We emphasize that our method is a **training objective on top of any linear-bounds method**. We will explicitly discuss **IBP and CROWN-IBP** as looser but cheaper alternatives that can plug into the same margin-over-slope formula when scalability to very deep networks is needed.
> - For cross-norm behavior, we will add a short section showing how the closed-form radius generalizes from $\ell_\infty$ to general $\ell_p$ via **dual norms**, while acknowledging that our current experiments only implement $\ell_\infty$.

---

> ### Author Response · Authors · 2025-12-03
>
> #### (d) Experiments and missing baselines
>
> Reviewers noted:
>
> - missing baselines (e.g., TRADES, recent certified training methods),
> - small MNIST network and small epsilon compared to state-of-the-art,
> - lack of baselines for the regression example, and
> - fairness concerns around PGD vs smoothing (different norms).
>
> **Our response and planned revision**
>
> - We **reframe the empirical scope** as proof-of-concept, not as a comprehensive benchmark against all recent defenses, and soften any language suggesting otherwise.
> - On MNIST, within our compute and page budget we will:
>   - add a **CROWN-based certified-loss baseline** (training on a standard adversarial cross-entropy upper bound from the same CROWN relaxations) to isolate the effect of the margin-over-slope objective, and
>   - add at least one additional adversarial-training baseline such as **TRADES** on the same architecture and epsilon.
> - For fairness:
>   - we will clearly state the **norm and epsilon used for each baseline**, and ensure that when we compare certified accuracies we use the **same norm and epsilon** across methods.
>   - randomized smoothing will be explicitly labeled as a **cross-norm reference** (an $\ell_2$ certified method included for context), not a directly comparable $\ell_\infty$-certified baseline.
> - For regression, we add comparisons to **natural training** and **attack-based robust training** (PGD on the regression loss) to show how certificate-driven training changes safety vs accuracy, and we will clarify that prior certified-training baselines for this specific setup are not available.
>
> If the situation permits, we will also include a **large CNN** on MNIST to demonstrate that the objective continues to behave well for large models.
>
> #### (e) Geometric claims and wording (over-selling “geometry” and differentiability)
>
> Reviewer F3jr in particular felt that:
>
> - we repeat geometric slogans without enough justification,
> - we overstate statements such as “adversarial training just moves boundaries” or “transforming expensive post-hoc verification”, and
> - we are imprecise about differentiability for ReLU networks and about what “geometry” CROWN actually exposes.
>
> **Our response and planned revision**
>
> - We will **remove or soften** claims that suggest adversarial training leaves geometry unchanged. Our intended point is that adversarial training does not explicitly optimize a certified radius, whereas our objective does; we will state this more carefully.
> - We clarify that we do **not** accelerate branch-and-bound; instead, we **avoid calling complete verifiers during training** by extracting a closed-form lower bound from CROWN’s linear relaxations. We will rewrite the relevant sentences accordingly.
> - We explain that by “geometric structure” we mean the **geometry of the certified outer region** induced by CROWN’s half-spaces (distances in input space to those planes), not the exact decision boundary of the network.
> - We make all differentiability statements precise: ReLU networks and CROWN bounds are **piecewise linear and differentiable almost everywhere**, and we rely on subgradient-based optimization as is standard. We will correct the wording in the introduction and theory sections.
> - We will also streamline the writing (remove repeated phrases) and clarify our disclosed use of language tools as **polishing**, not idea generation.
>
> #### (f) Ablations and component-wise contributions
>
> Reviewers ZNjT and others asked how much each part of the objective contributes:
>
> - margin vs slope,
> - smooth aggregation over constraints,
> - beta-CROWN tightening,
> - and training cost vs input dimension / number of constraints.
>
> **Our response and planned revision**
>
> We commit to add ablations that:
>
> - compare **margin-only**, **slope-only**, and **full margin-over-slope** objectives, reporting effects on clean accuracy and certified radius;
> - toggle **smooth aggregation** and **beta-CROWN joint tightening**, reporting their impact on training stability and robustness; and
> - present a small **scaling table** of per-epoch training time vs input dimension / constraint count on our regression instances.
>
> These analyses will clarify which components are essential and what the computational trade-offs are.
>
> #### (g) Scope: ReLU vs non-ReLU activations
>
> Several comments touched on the differentiability of the “slope” term for non-ReLU activations.
>
> **Our response and planned revision**
>
> - We emphasize that **all experiments use ReLU networks**. For this case, CROWN’s affine forms and our slope term are piecewise linear and differentiable almost everywhere.
> - We will **remove or soften** any claims suggesting that our current implementation directly handles general smooth activations such as sigmoid or tanh, and explicitly list extension to non-ReLU architectures as future work.

---

> ### Author Response · Authors · 2025-12-03
>
> ### 3. Overall
>
> In summary, the reviews converge on a common picture:
>
> - The **core idea**—turning CROWN’s linear relaxations into a **closed-form, differentiable certified-radius objective** and using it in hybrid training—is seen as sound and interesting.
> - The main concerns are about **positioning and scope**: clarifying novelty relative to PER and recent margin/Lipschitz methods, being precise about the validity and tightness of the bound, expanding discussion of complexity and scalability, and better aligning the experiments and baselines with current literature standards.
>
> Our rebuttal and planned revisions directly target these points with:
>
> - added comparisons and citations to PER, Fazlyab et al., Xu et al., Small Boxes, expressive-loss, and Lipschitz-network work;
> - explicit domain assumptions and tightness comparisons for the radius bound;
> - clearer complexity/scalability discussion and ablations on the components of the objective; and
> - additional baselines and clarifications in both classification and regression experiments.
>
> We hope this meta-response helps you see how the paper will evolve into a clearer and more tightly scoped contribution that connects verification and training, and we respectfully ask you to take these clarifications and planned improvements into account when making your decision.

---

### Meta-Review · Area_Chair_GwYR · 2025-12-18

**Summary:**

This work develops a certified robustness training method from a geometric perspective by calculating and optimizing input radius bounds from CROWN’s linear bounds, with a margin-over-slope objective that jointly enlarges safety margins and suppresses input sensitivity. Experimental results on MNIST classification and DC optimal power flow regression are presented.

The method looks very similar to a previous work “Training Provably Robust Models by Polyhedral Envelope Regularization”. Experiments are very preliminary, have setting issues, and do not contain any comparison against many previous works from the deterministic certified training literature. The writing of the paper has severe flaws. The authors promised many revisions which have not been added and will require a thorough review. As such, this paper should be rejected.

**Reviewer Concerns:**

There is no empirical comparison against any baseline from the deterministic certified training literature, and the currently available experiments are very preliminary and have setting issues.

The proposed method is technically similar to "Training Provably Robust Models by Polyhedral Envelope Regularization" and relevant to some others. Although methodological differences have been explained in the rebuttal, an empirical comparison is lacking for verifying the contribution of the proposed version.

One reviewer raised an important concern about the practical technicality of the lower bound which depends on intermediate pre-activation bounds computed for the entire given Linf ball. The author's response has partially addressed this, but empirical evidence is still lacking.

Concerns regarding the potential significant use of LLMs in writing the paper have been raised. Additionally, the paper contains several apparently LLM hallucinated references, including those for CROWN.

All above are outstanding concerns. There are other concerns like the cost of using CROWN, differentiability of bounds, overclaims about geometrical things, which are relatively minor and partially addressed by the additional explanation.

**Reviewer Scores:**

Reviewer JLtf: The major concern on the lack of comparison with previous works has not been resolved. Thus this reviewer is unlikely to change their score of 2.

Reviewer zNJt: This is the only reviewer who initially gave a positive rating of 6, and did not have major concerns. The reviewer may maintain their score or decrease the score upon considering other reviews.

Reviewer Sp2K: The concern on experiments remains. Thus this reviewer is unlikely to change their score of 2.

Reviewer F3jr: Major concerns remain but some questions have been answered or the authors have promised to soften their wording. This reviewer may raise their rating from 0 to 2, or maintain the original rating of 0.

---

### Decision · Program_Chairs · 2026-01-26

Reject